# Structural basis for human mitochondrial tRNA maturation

Vincent Meynier[1], Steven W. Hardwick[2], Marjorie Catala[1], Johann J. Roske[2], Stephanie Oerum[1], Dimitri Y. Chirgadze[2], Pierre Barraud[1], Wyatt W. Yue[3,4], Ben F. Luisi[2] & Carine Tisné[1] ✉

The human mitochondrial genome is transcribed into two RNAs, containing mRNAs, rRNAs and tRNAs, all dedicated to produce essential proteins of the respiratory chain. The precise excision of tRNAs by the mitochondrial endoribonucleases (mt-RNase), P and Z, releases all RNA species from the two RNA transcripts. The tRNAs then undergo 3′-CCA addition. In metazoan mitochondria, RNase P is a multi-enzyme assembly that comprises the endoribonuclease PRORP and a tRNA methyltransferase subcomplex. The requirement for this tRNA methyltransferase subcomplex for mt-RNase P cleavage activity, as well as the mechanisms of pre-tRNA 3′-cleavage and 3′-CCA addition, are still poorly understood. Here, we report cryo-EM structures that visualise four steps of mitochondrial tRNA maturation: 5′ and 3′ tRNA-end processing, methylation and 3′-CCA addition, and explain the defined sequential order of the tRNA processing steps. The methyltransferase subcomplex recognises the pre-tRNA in a distinct mode that can support tRNA-end processing and 3′-CCA addition, likely resulting from an evolutionary adaptation of mitochondrial tRNA maturation complexes to the structurally-fragile mitochondrial tRNAs. This subcomplex can also ensure a tRNA-folding quality-control checkpoint before the sequential docking of the maturation enzymes. Altogether, our study provides detailed molecular insight into RNA-transcript processing and tRNA maturation in human mitochondria.

Mitochondria generate most of the cellular energy in eukaryotes through oxidative phosphorylation. In human mitochondria, this capacity is critically dependent on the biogenesis within the organelle of 13 essential proteins of the respiratory chain that are encoded in the mitochondrial DNA genome (mt-DNA). The mt-DNA also encodes the RNA components of the specialised translation machinery of the organelle: 2 ribosomal RNAs (mt-rRNAs) and 22 transfer RNAs (mt-tRNAs)[1]. Each mt-DNA strand is transcribed as a large polycistronic RNA transcript containing all the RNA species required for mitochondrial translation[2–4]. Most of the individual RNA species are released by endonucleolytic excision of tRNAs from each polycistronic RNA transcript[1,3,5], constituting a key step of mitochondrial gene expression. These processing events likely occur co-transcriptionally[6,7] and are performed in RNA granules close to the mitochondrial nucleoid[8–11]. Processed mt-tRNAs further undergo the addition of the 3′-CCA trinucleotide and chemical modifications to become fully mature[12]. Impaired processing of mt-tRNAs has a detrimental effect on gene expression and ribosome assembly in the organelle[6,13], and mutations that disturb any stage of mt-tRNA maturation or mt-tRNA structures are linked to severe human diseases[14–16].

[1]Expression Génétique Microbienne, Université Paris Cité, CNRS, Institut de Biologie Physico-Chimique (IBPC), 75005 Paris, France. [2]Department of Biochemistry, University of Cambridge, Tennis Court Road, Cambridge CB2 1GA, UK. [3]Centre for Medicines Discovery, Nuffield Department of Clinical Medicine, University of Oxford, Oxford OX3 7DQ, UK. [4]Present address: Biosciences Institute, Newcastle University, Newcastle upon Tyne NE2 4HH, UK. ✉e-mail: carine.tisne@cnrs.fr

The mt-tRNA 5′-leader and 3′-trailer processing reactions are performed by the mt-RNase P and mt-RNase Z, respectively. In metazoan mitochondria, the endoribonuclease PRORP (protein-only RNase P) acts in place of the catalytic RNA component employed by the prototypical RNase P ribozyme[17–19], while the endoribonuclease ELAC2, a long-form RNase Z found only in eukaryotes, processes the 3′-trailer of both mitochondrial and cytosolic tRNAs[20,21]. The processing is ordered, with 5′-leader pre-RNA cleavage occurring prior to 3′-trailer processing[6,13,21,22]. A few mt-pre-tRNAs are organised as clusters such that the 3′-trailer of one tRNA borders the 5′-end of its neighbour. In the case of the tRNA$^{His}$-tRNA$^{Ser}$(AGY) pair, where tRNA$^{Ser}$(AGY) lacks the D-arm and has an extended anticodon stem, the processing of the 3′-trailer of tRNA$^{His}$ by ELAC2 simultaneously generates a mature 5′-end tRNA$^{Ser}$(AGY), without the need for PRORP[23].

Metazoan mt-RNase P is a multi-enzyme assembly comprising the endonuclease PRORP and a tRNA methyltransferase (MTase) subcomplex composed of the MTase, TRMT10C, and the dehydrogenase, SDR5C1[17,24,25]. These enzymes, but not their enzymatic activities, are strictly required for efficient tRNA cleavage by PRORP in vivo[17]. TRMT10C belongs to the SPOUT-fold Trm10 family of tRNA MTases found in archaea and eukaryotes. It catalyses the methylation of nitrogen-1 ($N^1$) of purine at position 9 of mt-tRNAs using S-adenosyl-L-methionine (SAM) as the methyl donor cofactor[24,26,27]. This dual specificity, i.e. the ability to methylate either nucleotides A9 or G9 in mt-tRNAs, is not shared by all Trm10-family enzymes[28]. So far, TRMT10C is also the only member of the Trm10 family that is active in complex with a protein partner. The $N^1$-methylation of purine-9 is the most prevalent nucleotide modification in human mt-tRNAs. Of the 22 mt-tRNAs, 19 have an N1-methylated purine 9, with 14 occurring on an adenine (m$^1$A) and 5 on a guanine (m$^1$G)[12]. The mt-tRNAs have low structural stability compared to cytosolic tRNAs because they have a low GC content and their D-, T- and variable loops that support the folding of cytosolic tRNAs[29], are either absent or differ in length. Nucleotide modifications are crucial to ensure functional folding and stability of these structurally divergent tRNAs required for mitochondrial translation. After mt-tRNA processing, the addition of 3′-CCA, which is required for aminoacylation, is catalysed without the requirement for a nucleic-acid template by TRNT1, a class-II nucleotidyltransferase[30,31].

In vitro, data suggest that the TRMT10C/SDR5C1 subcomplex may act as a maturation platform to facilitate sequential processing steps of the low-stability mitochondrial pre-tRNAs[27]. The pre-tRNA is bound to TRMT10C/SDR5C1 for purine-9 $N^1$-methylation and 5′-leader processing by PRORP, and remains associated to TRMT10C/SDR5C1 during subsequent maturation steps, i.e. 3′-processing by ELAC2 and 3′-CCA addition by TRNT1[27]. In vitro, the presence of the pre-tRNA in the TRMT10C/SDR5C1 subcomplex dramatically enhances the activity of PRORP[32], but has no impact on TRNT1 activity[27]. Distinct results have been obtained for ELAC2 activity in the presence of the TRMT10C/SDR5C1 subcomplex: either a significant enhancement of the ELAC2 activity for most mt-tRNAs[27] or a decrease in ELAC2 activity[32]. In vitro, the tRNA is not released from the TRMT10C/SDR5C1 subcomplex after 5′-cleavage or methylation and is thus only accessible in the subcomplex[27,32]. In vivo, the interaction of TRMT10C/SDR5C1 with ELAC2 and TRNT1 enzyme is supported by their localization in mitochondrial RNA granules[9,11,33], and by the interaction of TRMT10C with ELAC2 and with TRNT1 in affinity-capture mass spectrometry[34]. ELAC2 and TRNT1 also mature nuclear tRNA where TRMT10C/SDR5C1 is absent, implying that they are active without the TRMT10C/SDR5C1 subcomplex to process more structurally-stable cytosolic tRNAs.

The cryo-electron microscopy (cryo-EM) structure of mt-RNase P bound to the pre-tRNA$^{Tyr}$[25] has elucidated the organisation of this complex. A tetramer of SDR5C1 interacts with a monomer of TRMT10C that wraps around the mt pre-tRNA$^{Tyr}$, with PRORP poised to process

the pre-tRNA 5′-leader (PDB 7ONU)[25]. The requirement of the TRMT10C/SDR5C1 subcomplex for PRORP activity still remains unclear. Based on structural data, it was proposed that PRORP adopts an active conformation upon binding to the TRMT10C/SDR5C1-bound pre-tRNA[25]. However, PRORP has been recently shown to be active alone in vitro, the main function of TRMT10C/SDR5C1 probably being to direct PRORP's nuclease domain to the cleavage site, thereby increasing the rate and accuracy of cleavage[32]. The organisation of the mt-RNase P exposes the 3′-cleavage site of pre-tRNA, where ELAC2 may bind. However, due to a lack of structural data on ELAC2, it is not clear whether ELAC2 could bind the pre-tRNA simultaneously with PRORP or would require the enzymes to exchange. Moreover, no experimental structure is available for ELAC2 or for a eukaryotic tRNA-bound RNase Z. Therefore, the mechanisms of 3′-cleavage and how the hierarchy of processing reactions is ensured remain to be determined. Similarly, no structural information is available for tRNA recognition by TRNT1 and the mechanism of 3′-CCA addition.

To investigate the mechanism of mitochondrial pre-tRNA 3′-processing and 3′-CCA addition, and the potential role of TRMT10C/SDR5C1 as a tRNA-maturation platform, we reconstituted functional complexes containing human ELAC2 or TRNT1 enzymes with a mitochondrial pre-tRNA bound to TRMT10C/SDR5C1 complex. To investigate the dual specificity of TRMT10C, the methylation complexes were formed with pre-tRNAs containing either an A9 (pre-tRNA$^{His}$) or a G9 (pre-tRNA$^{Ile}$) and with S-adenosyl-L-homocysteine (SAH), the cofactor-product for methylation, bound in the TRMT10C active site. To visualise the processing of the tRNA$^{His}$-tRNA$^{Ser}$(AGY) pair, we reconstituted the mt-RNase P and mt-RNase Z complexes with this pair of pre-tRNAs. We determined the structures of these four mt-tRNA maturation complexes using single-particle cryo-EM at a global resolution ranging from 2.7 to 3.2 Å, revealing snapshots of the four tRNA maturation steps, i.e. $N^1$-methylation of purine 9, 5′-leader and 3′-trailer processing and 3′-CCA addition. These structures provide the molecular basis for the first steps of RNA-transcript processing in human mitochondria and for a chain of maturation events that takes pre-tRNA from its transcript state to a maturation state ready to be aminoacylated for mitochondrial translation. They rationalise the role of TRMT10C/SDR5C1 complex as a scaffold that supports pre-tRNA maturation for structurally-degenerate mitochondrial tRNAs.

## Results

### Structure determination of the human mitochondrial tRNA maturation complexes

To reconstitute human mt-tRNA maturation complexes, we recombinantly expressed in *E. coli* and purified the TRMT10C/SDR5C1 subcomplex, PRORP, ELAC2, TRNT1, and catalytically inactive mutants of the endoribonucleases, PRORP$_{D479A}$ and ELAC2$_{H548A}$. These mutations target catalytic residues that coordinate metal ions in the active site of these two families of endoribonucleases[35,36]. All proteins were expressed without the mitochondrial transport signal (MTS). The pre-tRNA substrates, pre-tRNA$^{Ile}$ and pre-tRNA$^{His}$-tRNA$^{Ser}$(AGY) pair (hereafter named pre-tRNA$^{His-Ser}$), were produced by in vitro transcription (Fig. 1a, b). pre-tRNA$^{His(0,Ser)}$ and pre-tRNA$^{Ile(0,4)}$, both with no 5′-leader, and tRNA$^{Ile}$ with no extensions were also produced to provide substrates for ELAC2 and TRNT1, respectively. In pre-tRNA$^{His(0,Ser)}$ and pre-tRNA$^{Ile(0,4)}$, the 0 corresponds to 0 nucleotide (nt) in the 5′-leader and, Ser and 4 corresponds to tRNA$^{Ser}$(AGY) and 4 nucleotides as 3′-trailer, respectively (Supplementary Fig. 1a). TRMT10C/SDR5C1 forms a stable complex with pre-tRNA, and addition of co-factors or co-factor analogues, i.e. SAH to TRMT10C, NADH to SDR5C1 and CTP, ATP or CDP to TRNT1 stabilised these enzymes (Supplementary Fig. 1b, c). Therefore, SAH, NADH and CDP were added for cryo-EM analysis. We performed methylation, processing and 3′-CCA addition assays to investigate whether reconstituted maturation complexes with pre-tRNA bound to TRMT10C/SDR5C1 complex are functional

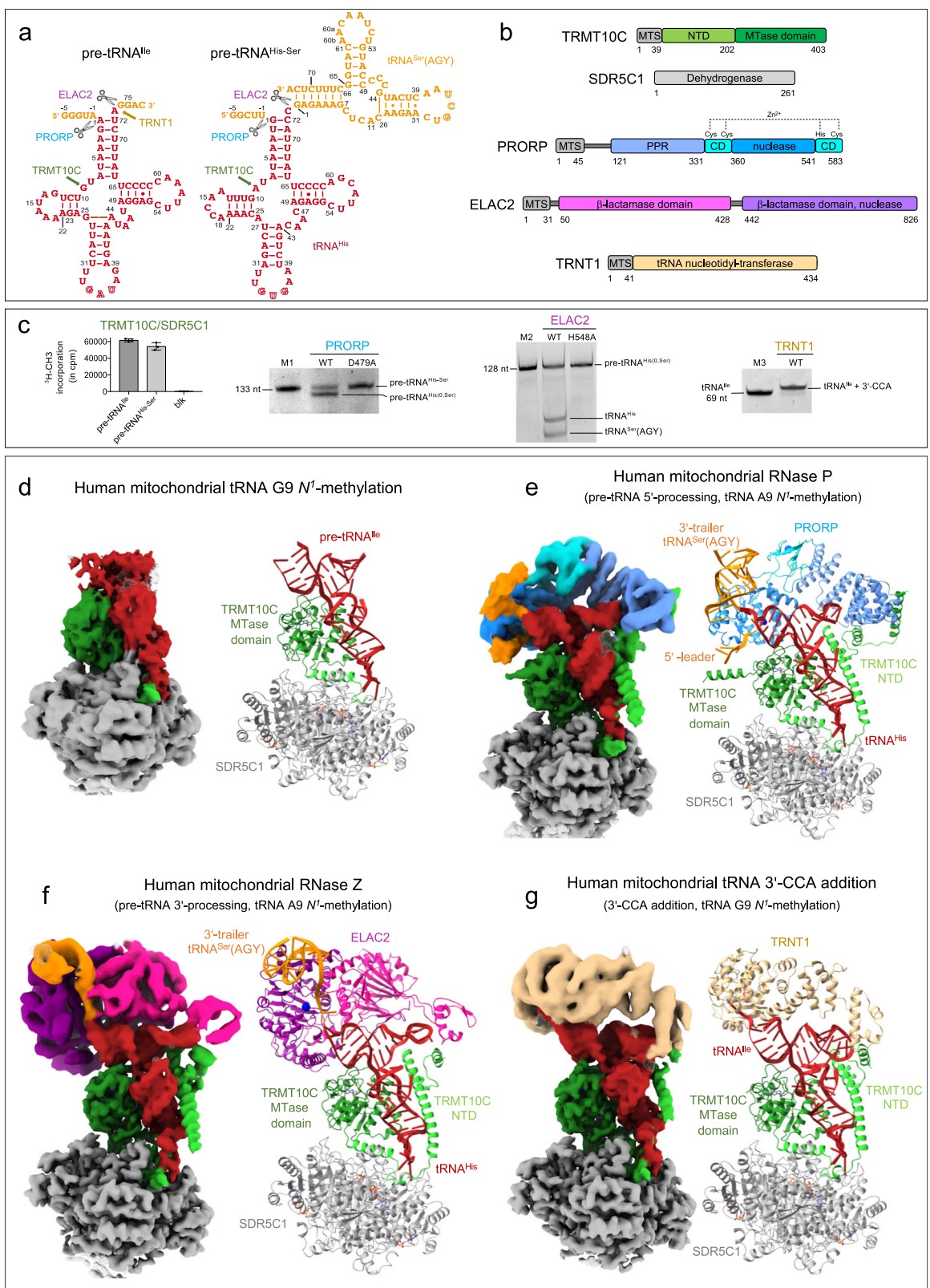

(Fig. 1c). Methylation assays, using [3]H-SAM as methyl donor, showed that the TRMT10C/SDR5C1 complex methylates both pre-tRNA[Ile] and pre-tRNA[His·Ser]. In pre-tRNA cleavage assays, PRORP processed pre-tRNA[His·Ser], whereas the PRORP[D479A] mutant did not. Pre-tRNA[His(0,Ser)] is a substrate of ELAC2, while the ELAC2[H548A] mutant was catalytically inactive. Lastly, in a 3′-CCA addition assay, TRNT1 could extend tRNA[Ile]

and CDP was used to block TRNT1 at the incorporation of the first C, as previously described[37].

For cryo-EM analysis, we first mixed TRMT10C/SDR5C1 with NADH and SAH, then added pre-tRNA, and finally either PRORP[D479A], ELAC2[H548A] or TRNT1 in excess. Single-particle cryo-EM data collection and processing yielded EM density maps at an overall resolution of

**Fig. 1 | Structures of the human mitochondrial tRNA maturation complexes.**
**a** Secondary structure of pre-tRNA$^{Ile}$ and pre-tRNA$^{His-Ser}$, the 5′-leader and 3′-trailer are indicated in orange, tRNA$^{Ser}$(AGY) is the 3′-trailer of tRNA$^{His}$. The sequence of the other pre-tRNAs used in this study and the nucleotide numbering used in the atomic coordinates deposited in the PDB are shown in Supplementary Fig. 1a. The maturation sites of the enzymes studied here are indicated. **b** Domain organisations of TRMT10C, SDR5C1, PRORP, ELAC2 and TRNT1; MTS: mitochondrial transport signal, NTD: N-terminal domain, MTase: methyltransferase, PPR: pentatrico-peptide repeat, CD: central domain. **c** Activity assays of reconstituted complexes showing methylation of pre-tRNA$^{Ile}$ and pre-tRNA$^{His-Ser}$ by the TRMT10C/SDR5C1 complex, processing of pre-tRNA$^{His-Ser}$ by PRORP, processing of pre-tRNA$^{His(0,Ser)}$ by ELAC2 and 3′-CCA addition to tRNA$^{Ile}$ by TRNT1 when pre-tRNAs are bound to the TRMT10C/SDR5C1 complex, PRORP$_{D479A}$ and ELAC2$_{H548A}$ are not active. pre-tRNA$^{His(0,Ser)}$ has no 5′-leader sequence, Blk is an assay without TRMT10C/SDR5C1.

Marker M1 is pre-tRNA$^{His-Ser}$ (133 nucleotides, nt), M2 is pre-tRNA$^{His(0,Ser)}$ (128 nt) and M3 is tRNA$^{Ile}$ (69 nt). Source data are provided as a Source Data file. For methylation assay, error bars represent standard deviation, $n = 3$ technical replicates, gels are representative of three experiments. **d** Cryo-EM density map and refined model of the complex responsible for human mitochondrial tRNA G9 $N^1$-methylation, composed of TRMT10C(SAH)/SDR5C1(NADH)/pre-tRNA$^{Ile}$ (Complex 1). **e** Composite map and refined model of the human mitochondrial RNase P with pre-tRNA$^{His-Ser}$, composed of TRMT10C(SAH)/SDR5C1(NADH)/pre-tRNA$^{His-Ser}$/PRORP$_{D479A}$ (Complex 2). **f** Composite map and refined model of the human mitochondrial RNase Z with pre-tRNA$^{His(0,Ser)}$, composed of TRMT10C(SAH)/SDR5C1(NADH)/pre-tRNA$^{His(0,Ser)}$/ELAC2$_{H548A}$ (Complex 3). **g** Composite map and refined model of the complex responsible for the human mitochondrial tRNA 3′-CCA addition, composed of TRMT10C(SAH)/SDR5C1(NADH)/RNA$^{Ile}$/TRNT1/CDP (Complex 4). Domains of enzymes are colored as in **b**.

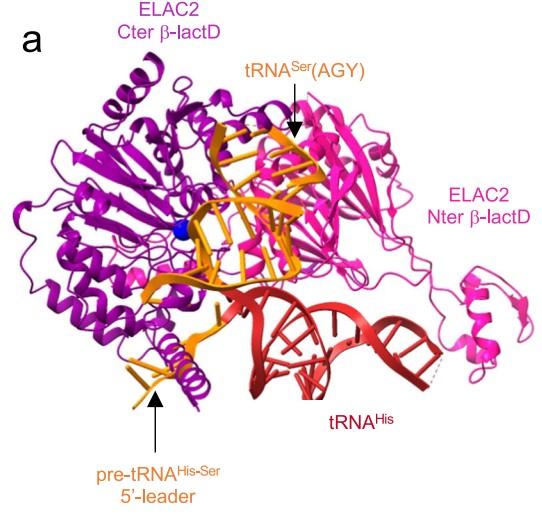

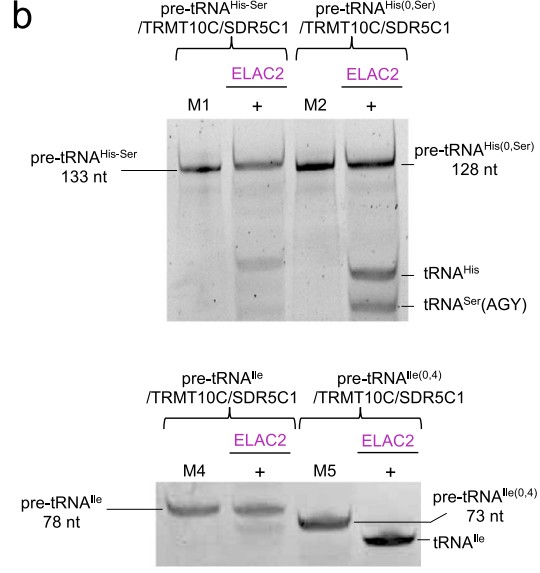

**Fig. 2 | ELAC2 cleavage is inhibited by the pre-tRNA 5′-leader. a** The pre-tRNA$^{His-Ser}$ from the mt-RNase P complex makes steric clashes with ELAC2 at the level of the pre-tRNA 5′-leader, indicated by the black arrow. **b** Cleavage assays by ELAC2 of pre-tRNA$^{His-Ser}$, pre-tRNA$^{His(0,Ser)}$, pre-tRNA$^{Ile}$ and pre-tRNA$^{Ile(0,4)}$ showing that ELAC2

cleavage is efficient on pre-tRNA without a 5′-leader. Markers M1, M2 are the same as in Fig. 1c, M4 is pre-tRNA$^{Ile}$ (78 nt) and M5 is pre-tRNA$^{Ile(0,4)}$ (73 nt). Gels are representative of three experiments, uncropped gels are in Source Data file.

3.2 Å for the TRMT10C/SDR5C1/pre-tRNA$^{Ile}$ complex, 2.7 Å for TRMT10C/SDR5C1/pre-tRNA$^{His-Ser}$/PRORP, 2.8 Å for TRMT10C/SDR5C1/pre-tRNA$^{His(0,Ser)}$/ELAC2 and 3.1 Å for TRMT10C/SDR5C1/tRNA$^{Ile}$/TRNT1 (Supplementary Figs 2-5; Supplementary Table 1). Focused refinements converged at overall resolutions of 4.5 Å for PRORP, 4.1 Å for ELAC2, and 4.5 Å for TRNT1. In all structures, a second copy of TRMT10C/pre-tRNA was bound on the opposite side of the SDR5C1 tetramer (Supplementary Fig. 6) as described for the RNase P/pre-tRNA$^{Tyr}$ complex[25], confirming the 2:4 TRMT10C:SDR5C1 stoichiometry previously reported[17,25,26]. The second copies displayed less clear densities, likely due to lower occupancy of TRMT10C, and we therefore focused our analysis on the better-resolved structural units. The resulting density maps allowed us to fit the crystal structure of TRMT10C with SAM (PDB 5NFJ)[26], the cryo-EM structure of mt-RNase P/pre-tRNA$^{Tyr}$ (PDB 7ONU)[25], the AlphaFold[38,39] models of the mt-tRNA maturation enzymes (Supplementary Table 1) and to model the pre-tRNAs including the tRNA$^{Ser}$(AGY) of the tRNA pair (Fig. 1). The density maps for the N-terminal domain (NTD) of TRMT10C were defined only for complexes where a maturation enzyme is bound on the pre-tRNA (Fig. 1d–g). Compared to the structure of mt-RNase P/pre-tRNA$^{Tyr}$ (PDB 7ONU)[25], we were able to build regions for this domain spanning from

residues 61–91, 157–174 and 386–403, to visualise the SAH in the TRMT10C active site and to build the 5′-leader and 3′-trailer of the pre-tRNA pair, resulting in a complete structure of the mt-RNase P.

## 5′-to-3′ order during mitochondrial tRNA-end processing

The architecture of the maturation complexes is globally the same in all structures, i.e. TRMT10C asymmetrically straddles a tetramer of SDR5C1, while wrapping around the pre-tRNA (Fig. 1e–g). Indeed, TRMT10C is anchored in the SDR5C1 tetramer via a central, mostly helical region that connects the TRMT10C MTase domain to its N-terminal domain (NTD). Both domains of TRMT10C bind an L-shaped pre-tRNA. This leaves one side of the pre-tRNA exposed for interaction with maturation enzymes. For the tRNA pair, tRNA$^{His}$ is bound in the TRMT10C/SDR5C1 subcomplex. The pre-tRNA binding sites of PRORP and ELAC2 overlap, excluding a simultaneous binding of PRORP and ELAC2 on the pre-tRNA and forcing the sequential action of those two nucleases. In our structure of mt-RNase P, PRORP interacts with the leader and the trailer sequences of tRNA$^{His}$ (Fig. 1e). In contrast, superimposition of the pre-tRNA$^{His-Ser}$ structure (with a 5′-leader) with that of pre-tRNA$^{His(0,Ser)}$ (with no 5′-leader) present in the complex with ELAC2 shows clashes between the 5′-leader of pre-tRNA$^{His-Ser}$ and ELAC2

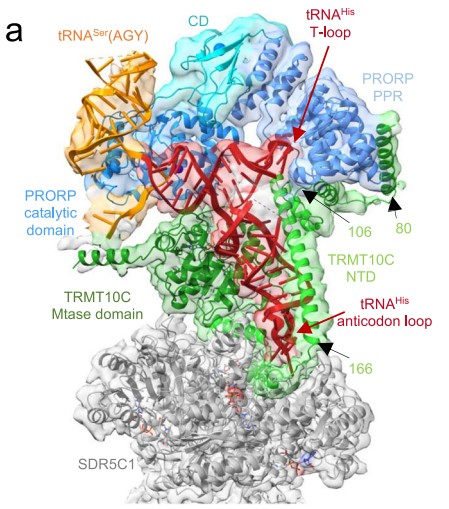
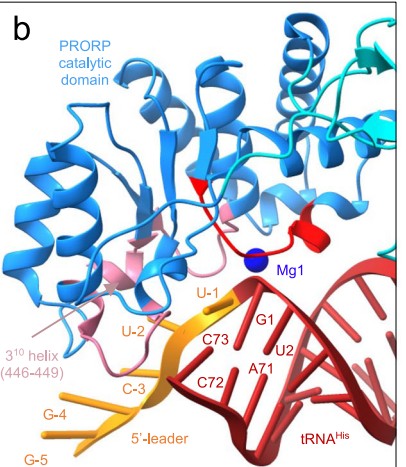
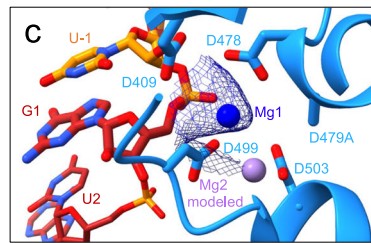
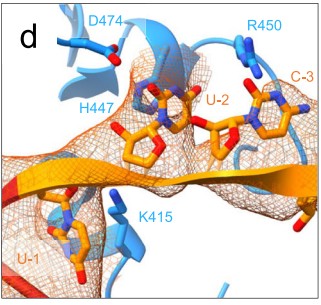
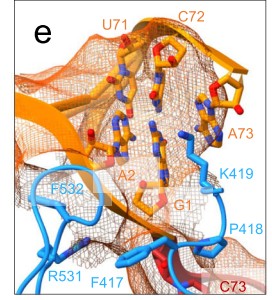
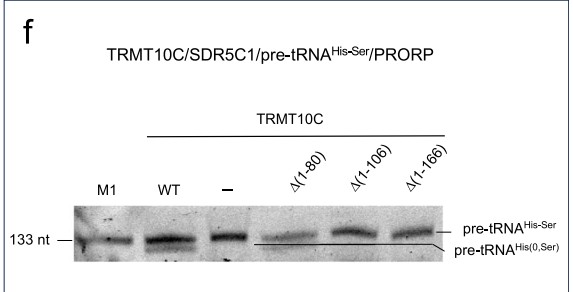

**Fig. 3 | Structure of the human mitochondrial RNase P complex processing pre-tRNA^His-Ser. a** Cryo-EM density map and cartoon representation of the structure of PRORP poised to process the TRMT10C/SDR5C1-bound pre-tRNA^His-Ser, with the color code indicated in Figure 1a, b. **b** Substrate-bound PRORP active site, in red loop interacting with the 5′-cleavage site and in pink, loops interacting with the 5′-leader of tRNA^His. **c** PRORP metal-ion binding in the active site. Residues in the active site and the pre-tRNA^His-Ser are shown in stick representation. Mg1 is shown as a marine sphere (with the density map in marine), and the Mg²⁺ ion labelled Mg2 as a light-purple sphere, was modeled on the basis of the PRORP1 crystal structure (PDB 4G24)[18]. **d** Interactions between the 5′-leader of pre-tRNA^His-Ser and PRORP. **e** Interactions between the 3′-cleavage site of pre-tRNA^His-Ser and PRORP. **f** Cleavage assays showing that PRORP activity is dependent on the interaction between PRORP and TRMT10C residues (1-106). Marker M1 is the same as in Fig. 1c, the gel is representative of three experiments, uncropped gel is in Source Data file. WT: TRMT10C/SDR5C1/PRORP/pre-tRNA^His-Ser, (-): PRORP/pre-tRNA^His-Ser, Δ(xx-yy): TRMT10C Δ(xx-yy)/SDR5C1/PRORP/ pre-tRNA^His-Ser, the location on the TRMT10C structure of the NTD residues 80, 106 and 166 are shown in **a**.

(Fig. 2a). These steric clashes occur at the level of the C-terminal helix of ELAC2 (residues 761-794) that is conserved in metazoans, present in *Saccharomyces cerevisiae*, but not in *Schizosaccharomyces pombe* and not conserved in plants (Supplementary Fig. 8a). These clashes likely prevent ELAC2 binding to a pre-tRNA with a 5′-leader. Indeed, the quantification of the band intensity of the cleavage product by ELAC2 (Fig. 2b) shows that only 20% of the cleavage products are observed when the 5′-leader is present on the pre-tRNAs, compared to 100% or 50% of cleavage products for pre-tRNA^Ile(0,4) and pre-tRNA^His(0,Ser), respectively, with no 5′-leader. ELAC2 cleavage activity is thus inhibited by the presence of the 5′-leader in pre-tRNA^Ile and pre-tRNA^His-Ser (Fig. 2b). This rationalises the 5′-to-3′ processing order previously observed for mt-pre-tRNA maturation in vivo[6].

**Structure of human mitochondrial RNase P processing pre-tRNA**
The cryo-EM structure of mt-RNase P in complex with pre-tRNA^His-Ser shows PRORP bound to the acceptor- and T-arms of pre-tRNA^His through its catalytic and PPR domains, respectively (Fig. 3a). The 5′-cleavage site of pre-tRNA^His is positioned in the catalytic center of PRORP (Fig. 3b). For this, a stretch of conserved residues in PRORP (residues 497-504, in red on Fig. 3b), that includes the conserved aspartates D499 and D503 involved in coordinating one of the two catalytic Mg²⁺ ions used in this family of enzymes for RNA cleavage[18], interacts with the phosphodiester backbone of nucleotides G1 and U2

of tRNA^His (Fig. 3c). In addition, nucleotides U-1, U-2 and C-3 of the tRNA^His 5′-leader are in contact with the catalytic domain of PRORP through two loops (residues 415-422 and 473-478) and a 3₁₀ helix, a helix with three residues per turn, formed by residues 446-449, that contain residues conserved in mammals (in pink on Fig. 3b, Supplementary Fig. 7a). For instance, H447 is likely π-stacked on U-2 (Fig. 3d). The Mg²⁺ ions are proposed to be coordinated by D479/D499/D503 and D409/D478/D479 (mutated to alanine here), respectively (Fig. 3c). The density map is consistent with one Mg²⁺ ion coordinated by the scissile phosphate, D499 and D478 (Fig. 3c). The second ion is not observed due to the D479A mutation, but could be modelled using the crystal structure of PRORP from *A. thaliana*[18]. PRORP is anchored to the pre-tRNA by interactions with both the 5′- and 3′- cleavage sites. Indeed, the density map also reveals interactions between the phosphodiester backbone of the nucleotides of the 3′-cleavage site, C73 of tRNA^His together with G1 and A2 of tRNA^Ser(AGY), and two long loops of PRORP that contains conserved residues (e.g. F417, P418, K419 and F532; Fig. 3e). These interactions likely block the stacking of the acceptor stems of the two tRNAs, that would prevent 5′-processing by PRORP.

Two sites of interaction between TRMT10C and PRORP were revealed in our structure. First, the density map suggests an interaction between PRORP (residues R384-K385) and the catalytic loop of TRMT10C (residues 310-320). The SAM cofactor and the pre-tRNA

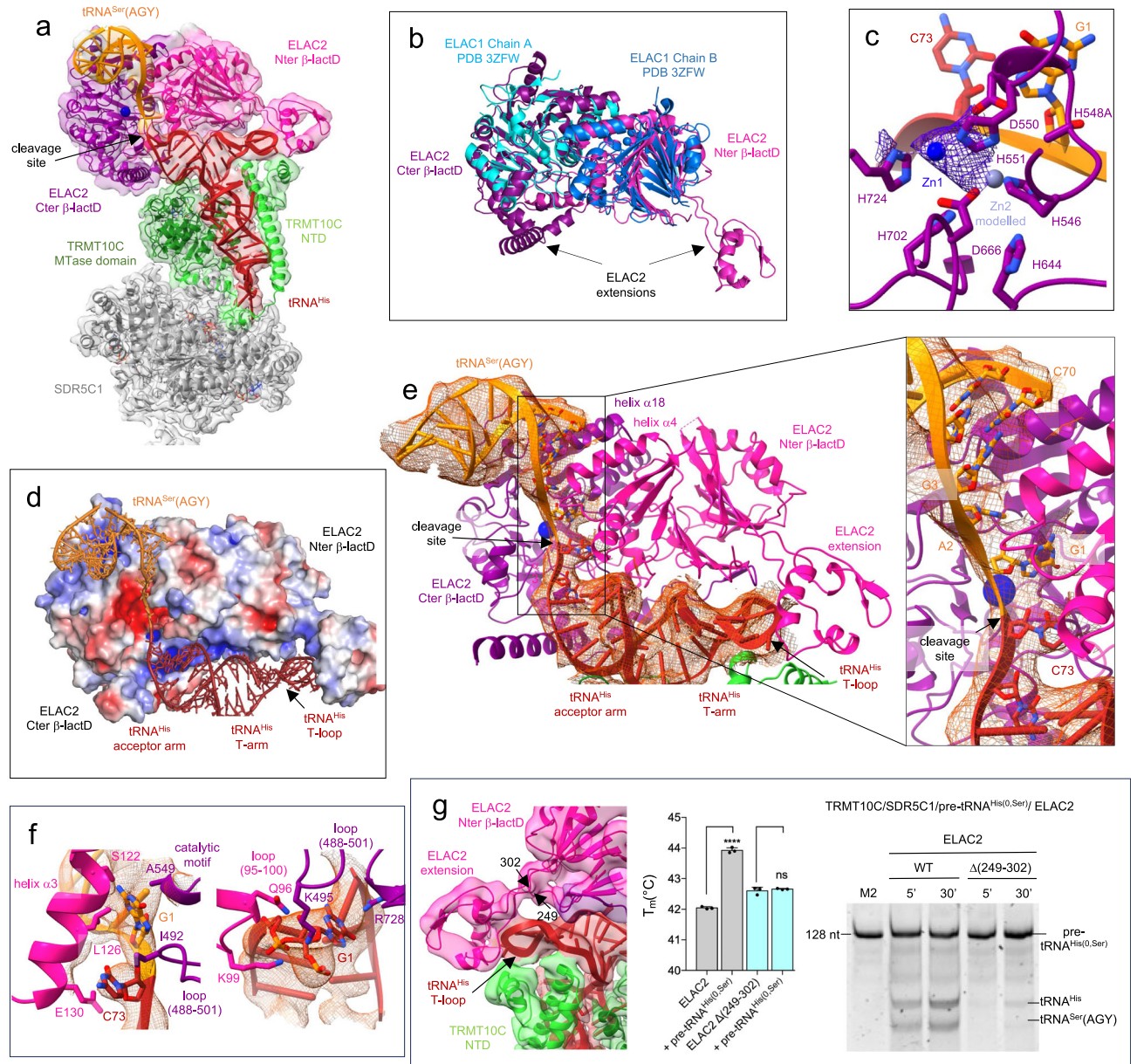

**Fig. 4 | Structure of the human mitochondrial RNase Z in pre-cleavage complex with pre-tRNA$^{His(0,Ser)}$.** **a** Cryo-EM density map and cartoon representation of the structure of ELAC2 poised to process the TRMT10C/SDR5C1-bound pre-tRNA$^{His(0,Ser)}$, with the color code indicated in Fig. 1a, b. **b** Superimposition of ELAC2 with ELAC1 homodimer (PDB 3ZWF). **c** Substrate-bound ELAC2 active site and metal-ion binding. Residues in the active site and the pre-tRNA$^{His(0,Ser)}$ are shown in stick representation. Zn1 is shown as a solid marine sphere (with the density map in purple), and Zn2 was modeled on the basis of the ELAC1 crystal structure (PDB 3ZWF), and is shown as a light-purple sphere. **d** Vacuum electrostatics surface representation of ELAC2 in the complex with TRMT10C/SDR5C1-bound pre-tRNA$^{His(0,Ser)}$. Positive charges are in blue, whereas negative ones are in red with the maximum color saturation corresponding to −5 kT/e (red) and +5 kT/e (blue). pre-tRNA$^{His(0,Ser)}$ is represented as ribbon and sticks with the color code indicated in Fig. 1a. **e** Substrate-bound ELAC2 active site and pre-tRNA$^{His(0,Ser)}$ binding, the insets shows the binding of C73 from tRNA$^{His}$ and, G1 and A2 from tRNA$^{Ser}$(AGY) at the interface between the ELAC2 β-lactamase domains with the density map for the tRNAs in orange. **f** Interactions between ELAC2 and, C73 from tRNA$^{His}$ and G1 from tRNA$^{Ser}$(AGY); Interactions between ELAC2 and G1 from tRNA$^{His}$; the density map is represented in orange for the tRNAs. **g** Density map showing the interactions between ELAC2 and TRMT10C at the level of the T-loop of tRNA$^{His}$; DSF assays testing the ability of ELAC2 and deletion mutant Δ(249-302) to bind pre-tRNA$^{His(0,Ser)}$. Data were analysed by one-way ANOVA using a F test statistic. P-values are indicated as follows: ns for $P = 0.61$ and **** for $P < 0.0001$, $n = 3$ technical replicates; the error bars are SD. Cleavage assays of ELAC2 and deletion mutant Δ(249-302) on pre-tRNA$^{His(0,Ser)}$ for 5 and 30 minutes, marker M2 is as in Fig. 1c. The gel is representative of three experiments. Source data are provided as a Source Data file.

define this loop conformation (Supplementary Fig. 7b). The presence of SAM is not required for PRORP activity, but this interaction could explain why SAM binding increases the binding efficiency and 5′-end processing by PRORP for some mt pre-tRNAs, like pre-tRNA$^{Val}$[40]. Second, the PRORP PPR domain that interacts with the T-loop of pre-tRNA, further binds two helices of the TRMT10C NTD (residues 61-106) to

form an interaction surface for the tRNA T-loop (Fig. 3a). Notably, helix α1 of TRMT10C (residues 61-82), which was not observed previously[25], interacts with the PPR domain of PRORP. Deletion of this helix α1 has no apparent effect on PRORP activity, whereas once the two first helices of TRMT10C are deleted, i.e. in the TRMT10C Δ(1-106) and TRMT10C Δ(1-166) mutants, the TRMT10C/SDR5C1 complex is no

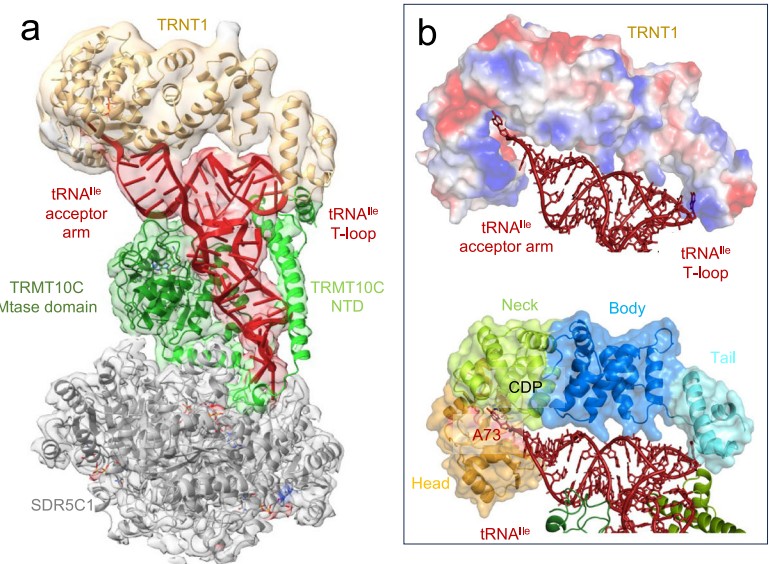

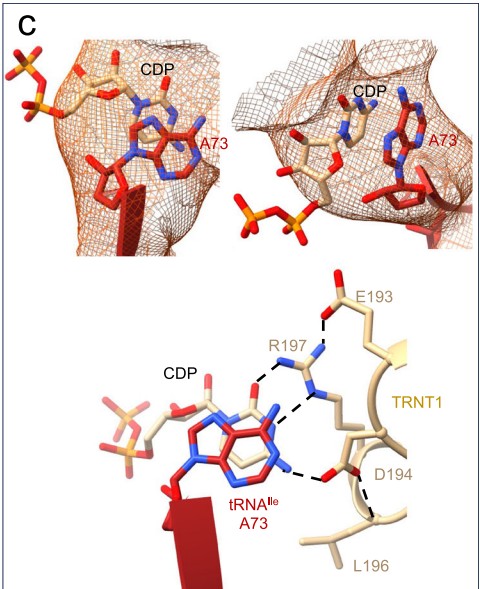

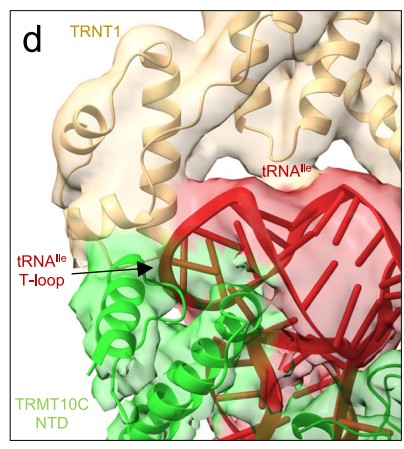

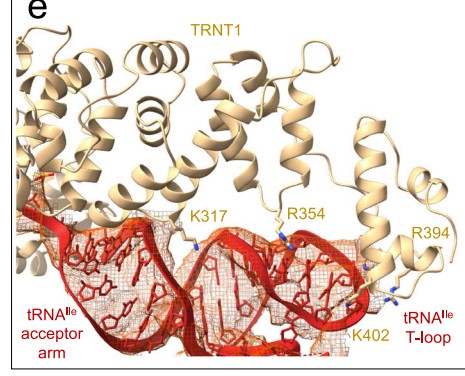

**Fig. 5 | Structure of the human mitochondrial CCA-adding enzyme maturing tRNA^Ile. a** Cryo-EM density map and cartoon representation of the structure of TRNT1 poised to add the first cytosine to tRNA^Ile, with the color code of Fig. 1a, b. **b** Vacuum electrostatics surface representation of TRNT1 in the TRMT10C(SAH)/SDR5C1(NADH)/tRNA^Ile/TRNT1 structure. Positive charges are in blue, whereas negative ones are in red with the maximum color saturation corresponding to −5 kT/e (red) and +5 kT/e (blue). tRNA^Ile is represented as red ribbon and sticks. The different domains of TRNT1 are indicated. **c** Cryo-EM density map showing the stacking of A73 of tRNA^Ile with the CDP in the TRNT1 active site and base-specific interactions between TRNT1 and CDP. Potential hydrogen bonds are shown as dashed black lines. The residues of the catalytic site are represented as beige sticks. **d**, Interaction between TRNT1 (in beige), the T-loop of tRNA^Ile (in red) and TRMT10C (in green). **e** Interactions between the body and tail domains of TRNT1 (in beige) and tRNA^Ile (in red).

longer able to promote PRORP-mediated cleavage of pre-tRNA^His-Ser (Fig. 3f). The TRMT10C (1-106)/PRORP interactions therefore clamp PRORP on the pre-tRNA.

## Structure of human mitochondrial RNase Z

The cryo-EM structure of the TRMT10C/SDR5C1/pre-tRNA^His(0,Ser)/ELAC2 complex reveals ELAC2 bound to pre-tRNA^His, poised to process the pre-tRNA^His 3′-trailer. The scissile phosphodiester bond between tRNA^His and tRNA^Ser(AGY) is actually positioned in the active site of ELAC2 (Fig. 4a). ELAC2 contains two β-lactamase domains that align well with the dimeric short-form human RNase Z (ELAC1, Fig. 4b). However, each β-lactamase (β-lactD) domain of ELAC2 has extensions from the core that are not present in ELAC1 and that are used to bind pre-tRNAs (Fig. 4b). As a member of the metallo-β-lactamase super-family, ELAC2 has a highly conserved metal-coordinating signature motif $^{546}$HxHxDH$^{551}$, together with H644, D666, H702 and H724, that coordinates two $Zn^{2+}$ ions to constitute the catalytic site (Fig. 4c; Supplementary Fig. 8a). The N-terminal β-lactamase domain (Nter β-

lactD, Fig. 4 in pink) lacks the residues coordinating $Zn^{2+}$ ions, and thus is catalytically inactive. The density map suggests the presence of only one of the two $Zn^{2+}$ ions, which is consistent with the H548A mutation (Fig. 4c). The second ion can be modelled from the structure of the human short-form RNase Z ELAC1 (PDB 3ZWF) (Fig. 4c). The structure therefore represents ELAC2 primed for cleavage, with the catalytic residues in position to coordinate two metal ions required for endo-nucleolytic cleavage and the scissile phosphodiester bond positioned in the active site (Fig. 4a, c).

tRNA^His adopts an L-shaped conformation and its four arms interact with TRMT10C, similarly to the complex with mt-RNase P. The acceptor and T-arms of pre-tRNA^His are involved in extensive contacts with ELAC2, mostly with the Nter β-lactD that offers a shape and charge-complementary surface to bind the pre-tRNA (Fig. 4d). Only the acceptor arm and the T-stem of tRNA^Ser(AGY) are visible in the density map (Fig. 4a, d). They are stacked on each other, forming a long, kinked RNA helix that interacts with the two β-lactD of ELAC2, in a positively-charged region (Fig. 4d) at the exit of the active site, formed by helix α4

(residues 142-155) of the Nter β-lactD and helix α18 (residues 582-594) of the Cter β-lactD (Fig. 4e). Nucleotides G1 and A2 of tRNA$^{Ser(AGY)}$ appear unpaired in the density map and are buried together with nucleotide C73 of tRNA$^{His}$ in the ELAC2 active site formed by the interaction of the two β-lactDs (Fig. 4e). The 3'-cleavage site nucleotides, C73 of tRNA$^{His}$ and G1 of tRNA$^{Ser(AGY)}$, are sandwiched between helix α3 (residues 121-131) of the ELAC2 Nter β-lactD, and a loop (residues 488-501) plus the catalytic motif $^{548}$HADH$^{551}$ of the ELAC2 Cter β-lactD (Fig. 4f). Moreover, nucleotide G1 of tRNA$^{His}$ interacts with a loop (residues 95-100) of the ELAC2 Nter β-lactD, and a loop (residues 488-501) plus the second catalytic motif (724-730) of the ELAC2 Cter β-lactD (Fig. 4f). Both β-lactDs thus participate in the interaction with the pre-tRNA pair at the level of the 3'-cleavage site. On the opposite side of pre-tRNA$^{His}$, the extension of the ELAC2 Nter β-lactD (residues 249-302), absent in ELAC1, interacts with the T-loop of pre-tRNA$^{His}$ (Fig. 4g). Deletion of the Nter β-lactD extension abolishes tRNA binding and thus inhibits cleavage activity (Fig. 4g), showing that this fragment of ELAC2 is a small RNA-binding domain that recognises the T-loop of mt-tRNAs and drives the tRNA-binding of ELAC2. The density map suggests an interaction between ELAC2 and the NTD domain of TRMT10C (Fig. 4g), but to a lesser extent than for PRORP. In the complex with ELAC2, the NTD domain of TRMT10C can be constructed from residue 91 in the density map, compared with residue 61 for the complex with PRORP. This suggests that TRMT10C NTD binds less strongly to ELAC2 than to PRORP. This is in agreement with the fact that the deletion of the TRMT10C NTD helices that interact with the pre-tRNA and/or with ELAC2 does not inhibit ELAC2 activity (Supplementary Fig. 8b). The interaction between TRMT10C and ELAC2 is thus not required for this pre-tRNA 3'-trailer processing, in contrast to what is observed for PRORP. The structure of human ELAC2 processing pre-tRNA$^{His(0,Ser)}$ reveals the structure of eukaryotic RNase Z bound to a substrate pre-tRNA.

### 3'-CCA addition by TRNT1

The cryo-EM structure of the TRMT10C/SDR5C1/tRNA$^{Ile}$/TRNT1 complex shows the TRNT1 enzyme bound to tRNA$^{Ile}$ for incorporation of the first cytosine of the CCA motif at the tRNA$^{Ile}$ 3'-end (Fig. 5a). TRNT1 adopts a sea-horse-shaped structure, providing a positively charged surface that binds the acceptor- and T-arms of the tRNA (Fig. 5b). The TRNT1 active site lies at the interface between two domains called head and neck in this family of proteins, and hosts the 3'-terminus of tRNA$^{Ile}$ (Fig. 5b) and a molecule of CDP that is likely π-stacked with the substrate A73 nucleotide (Fig. 5c). The head domain (residues 1-178) harbours conserved catalytic carboxylate residues, i.e. D77, D79 and E121 that coordinate two Mg$^{2+}$ ions in enzymes of the class-II nucleotidyl-transferase family[41]. Although density could not be observed for Mg$^{2+}$, two ions can be modelled into the active site of TRNT1 guided by the crystal structure of the *B. stearothermophilus* CCA-adding enzyme[42] (Supplementary Fig. 9a). The neck domain (residues 179-267) contains key nucleobase-interacting residues conserved in class-II enzymes[41]. Base-specific interactions can occur between the CDP and conserved residues D194 and R197, further stabilised by interactions with L196 and E193 (Fig. 5c, Supplementary Fig. 9b), as observed for the *B. stearothermophilus* CCA-adding enzyme[42]. This interaction pattern is equivalent to that made by cytosines in GC Watson-Crick base-pairing. D194 and R197 can also specifically accommodate an ATP molecule in the active site of TRNT1 (Supplementary Fig. 9c). This allows the selection of CTP or ATP without the need for a nucleic-acid template.

The body (residues 268-354) and tail (residues 354-434) domains of TRNT1 bind the backbone of the tRNA$^{Ile}$ acceptor- and T-arms (Fig. 5a, b). The tail domain of TRNT1 forms a shape-and-charge complementary surface that accommodates the T-loop of mt-tRNA$^{Ile}$, which could be entirely built in this complex (Fig. 5e). The density map shows that the T-loop backbone of tRNA$^{Ile}$ interacts with TRNT1 in a region where TRNT1 may also interact with TRMT10C (Fig. 5d). No

interaction is observed between the tRNA D-loop and TRNT1. The two related enzymes, *A. aeolicus* A-adding[43] and *T. maritima* CCA-adding[44], recognise both the elbow of tRNAs formed by the interaction between the D- and T-loops via their tail and body domains. TRNT1 therefore binds to mitochondrial pre-tRNAs using a different mode of recognition to that observed for other class-II nucleotidyltransferases.

### tRNA-binding by TRMT10C/SDR5C1 subcomplex

In all structures, the TRMT10C/SDR5C1 complex binds an L-shaped pre-tRNA, even if the elbow of the pre-tRNA is not formed (Fig. 1, Supplementary Fig. 10a). All pre-tRNA arms are engaged in extensive contacts with TRMT10C and to a lesser extent with SDR5C1 chain C at the level of the anticodon loop (Supplementary Fig. 10b). Both domains of TRMT10C participate in pre-tRNA binding. The NTD of TRMT10C loops around the tRNA, opposite the MTase domain, by surrounding the anticodon, D- and T-arms, and the variable loop (Fig. 6a). The MTase domain of TRMT10C makes extensive contacts with the backbone of the acceptor, D and anticodon arms of pre-tRNA substrates (Fig. 1, Fig. 6a).

The anticodon loop adopts the same distinctive conformation in all structures available with anticodon-loop bases of nucleotides 32, 34, 35, 36, and 38 being flipped out (Supplementary Fig. 10a). This conformation is stabilised by numerous interactions with TRMT10C through a stretch of amino acids (157-185), built entirely in the structures presented here (Fig. 6a), notably for residues 157-174 not visible in PDB 7ONU[25]. F177 π-stacks with U35 in tRNA$^{His}$ (A35 in tRNA$^{Ile}$). K164 and R157 interact with G36 that further π-stacks with A38. The conserved residue L180 interacts with G34 that is further stabilised by chain C of SDR5C1 via residues (98-105) from a lysine-rich loop that also interacts with U32 (Supplementary Fig. 10b). In addition, the anticodon loop binds to TRMT10C through interactions between the conserved residue D184 and nucleotide A37 of tRNA$^{His}$ that is also π-stacked with A31-U39 base pair and U33 (Fig. 6a). R181 protrudes into the anticodon loop and interacts with the O2 atom of U33, the nucleotide just upstream of the anticodon triplet. This position conserved as a pyrimidine in all mt-tRNAs (21 as a uridine and 1 as a cytidine for tRNA$^{Met}$) (Fig. 6a). R181 also contacts the phosphodiester backbone of G34 and A37 in tRNA$^{His}$. Compared to the TRMT10C MTase structure in complex with SAM[26] or with pre-tRNA[25], a conformational change is observed for the cofactor-binding loop (residues 310-320) and the RNA-binding loop (residues 342-352, Fig. 6b). The RNA-binding loop sits in a groove formed by the acceptor arm, and binds to nucleotides 67-69 in tRNA$^{His}$. These conformational changes of loops in the TRMT10C active site imply that it is necessary to determine structures of TRMT10C bound to the SAH and a substrate tRNA to visualise the pre-catalytic state of methylation.

One of the four α-helices in the TRMT10C NTD is very long (65 Å, residues 126-166) and contains conserved positively-charged (K131, K143, K149 and K150) and aromatic (Y132, Y135) residues that mediate extensive interactions with pre-tRNAs from their anticodon loop to their acceptor stem. This α-helix lies in a groove of the mt-pre-tRNA formed by the D-stem and likely stabilises the L-shaped conformation of the pre-tRNA in the absence of interaction between the D- and T-loops, that would otherwise form the elbow of a "regular" tRNA structure (Fig. 6c). All nucleotides of the variable loop (nucleotides 44-47) are in contact with this α-helix, e.g. A46 is flipped out and π-stacked with the conserved aromatic Y135 residue (Fig. 6c inset). A46 is further bound to K218 from the TRMT10C MTase domain (Fig. 6c). This helix is required for the binding of pre-tRNA, but not for the binding of SAM, and its deletion abolishes the MTase activity of TRMT10C (Fig. 6d, Supplementary Fig. 10 c, d, e).

### N$^1$-methylation of A9 or G9 in mt tRNA by TRMT10C

In all the structures, the TRMT10C catalytic pocket contains SAH and the flipped-out substrate base: nucleotide A9 of pre-tRNA$^{His-Ser}$ and

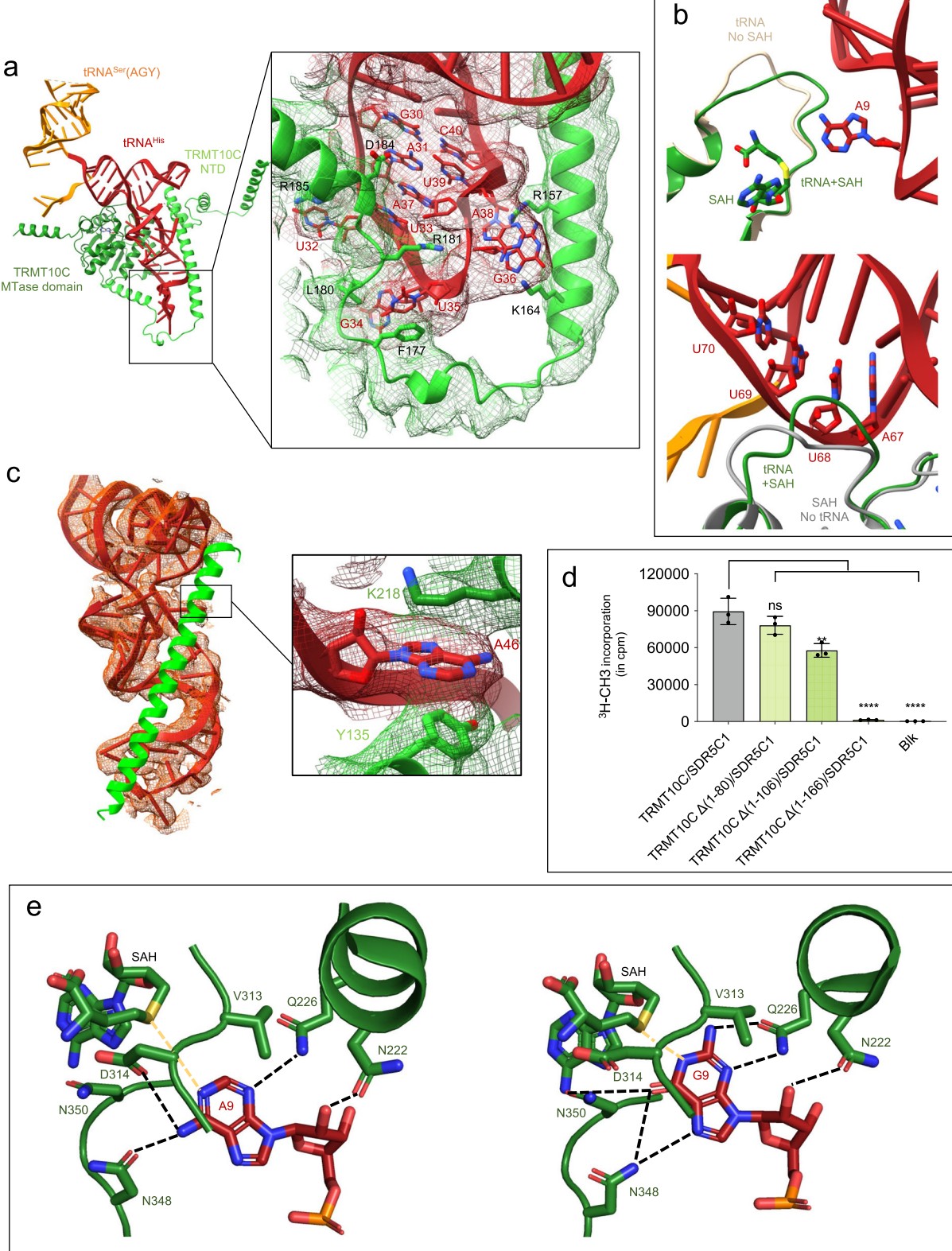

nucleotide G9 of pre-tRNA[Ile]. SAH is bound via interaction with the same set of conserved residues previously determined in the X-ray structure of TRMT10C MTase domain bound to SAM[26] (Supplementary Fig. 11). The flipped purine-9 is stacked against the conserved V313 and further stabilised in the TRMT10C active site by interaction with the same set of conserved residues, i.e. N222, Q226, D314 and N350 (Fig. 6e, Supplementary Fig. 11). Both m1G9 and m1A9 MTase activities

of TRMT10C are abolished with the Q226A or D314N substitution in TRMT10C[24,26], confirming the essential roles of these strictly invariant residues. Q226 interacts with the N3 atom of A9 and with the N3 and NH2 group of G9, which are key for the selection of purines over pyrimidines. N222 interacts with the 2′-OH of the ribose of nucleotide 9. On the opposite side of purine-9, N348 and N350 interact with O6 and N7 of G9. In the case of A9, N348 operates a conformational switch

**Fig. 6 | TRMT10C interactions with pre-tRNA for purine-9 $N^1$-methylation.**
**a** Overview of the TRMT10C/pre-tRNA$^{His-Ser}$ structure (Complex 2), SDR5C1 and PRORP have been removed for clarity. TRMT10C is represented as cartoon with the color code indicated in Fig. 1 and the pre-tRNA$^{His-Ser}$ as red ribbon and sticks. The inset shows a close-up of the anticodon loop and TRMT10C structure in the cryo-EM density map. **b** Conformation of the SAM/SAH-binding loop in the TRMT10C MTase domain upon binding of SAH and tRNA (complex 2, in green) compared to the conformation in the structure with tRNA and without SAM/SAH (PDB 7ONU, in beige)[25]; Conformation of the RNA-binding loop in the TRMT10C MTase domain upon binding of tRNA (complex 2, in green) compared to the conformation in the structure of TRMT10C with no tRNA, but with SAM (PDB 5NFJ, in grey).

**c** Interactions between tRNA$^{His}$ (in red) and the helix (126-166, in green) of TRMT10C, the inset shows the interaction of nucleotide A46 with Y135 from TRMT10C NTD and K218 from TRMT10C MTase domain. **d** Methyltransferase assays with TRMT10C with different N-terminal deletions, Blk: assay without pre-tRNA. Data were analysed by one-way ANOVA using a Dunnett's multiple comparison test. $P$-values are indicated as follows: ns for $P = 0.15$, *** for $P = 0.0004$ and **** for $P < 0.0001$, $n = 3$ technical replicates. The error bars are SD. Source data are provided as a Source Data file. **e** Interactions between A9 of pre-tRNA$^{His-Ser}$ and G9 of pre-tRNA$^{Ile}$ with conserved residues in the active site of TRMT10C. Potential hydrogen bonds are shown as dashed black lines.

enabling the interaction with the NH$_2$ group of A9 (Fig. 6e). This flip of the N348 side chain could participate in the dual specificity of TRMT10C. For the human TRMT10A and the *S. cerevisae* Trm10 enzymes that have a m$^1$G9 activity, the N348 position is occupied by R243 in the AlphaFold models of these enzymes (AF-Q12400 and AF-Q8TBZ6), where an arginine allows the accommodation of a guanine but not an adenine in the active site (Supplementary Fig. 11c).

Conserved residues of TRMT10C MTase domain (N222, E229, K315, D354, Q355, R379, K378) also contact the backbone of nucleotides 9, 10 and 11. Based on the position of SAH, a model with SAM can be generated to explain the reaction mechanism mediated by some specific residues. The transferrable methyl group of SAM is located 3.0 Å from the N1 atom of A9 and 1.7 Å from G9 in the TRMT10C active site. D314 is poised to form a hydrogen bond with the NH$_2$ group of A9. These interactions support an S$_N$2 mechanism for the methyl transfer from the SAM to the base. Although no residue is in an appropriate position to allow the deprotonation of the N$_1$ atom of G9, D314 appears to be the best catalytic candidate, and its mutation to alanine abolishes the TRMT10C $N^1$-methylation activity[26].

## Discussion

Here we have determined the structures of human mitochondrial RNase P and RNase Z, that process the primary mitochondrial RNA transcripts, to release the individual mitochondrial RNA species required for mitochondrial translation of the respiratory chain proteins encoded in the mitochondrial genome. We have also determined the structures of the human mitochondrial purine-9 tRNA methylation and 3′-CCA addition complexes, that mature mitochondrial tRNAs. The structures reveal how pre-tRNAs are recognised, cleaved, methylated and prepared for remaining steps of maturation (aminoacylation and nucleotide modifications) to be fully functional in mitochondrial translation.

The structures of human ELAC2 and TRNT1 reveal structures of eukaryotic RNase Z and nucleotidyltransferase, respectively, in complex with a substrate tRNA. They show the architecture of the complexes during catalysis and shed light on mechanisms of tRNA recognition and cleavage or 3′-CCA addition. In these structures, SAH is bound to the TRMT10C active site, revealing the substrate base (A9 or G9) of the tRNA flipped into the active site and implicating D314 as a general base catalyst for the S$_N$2 methylation reaction. The TRMT10C NTD domain acts as an RNA binding domain, and structural comparison of TRMT10C with human TRMT10A, TRMT10B and *S. cerevisiae* shows that their NTD adopts the same fold with a long α-helix that can interact from the anticodon loop to the T-arm of tRNA (Supplementary Fig. 12). For Trm10 enzymes from Archaea, this helix is not conserved and additional domains with unrelated folds are used to bind tRNAs (Supplementary Fig. 12).

Our structures explain how the hierarchy of the pre-tRNA processing reactions is ensured. First, the two processing enzymes cannot bind simultaneously to the pre-tRNA, so they must act sequentially, which necessitates an exchange of the endonuclease bound to the pre-tRNA. Second, the mitochondrial RNase Z cannot bind to a pre-tRNA containing a 5′-leader, whereas RNase P can. This imposes a 5′-to-3′

order for the pre-tRNA processing steps. The structure of mt-RNase P in complex with a pair of pre-tRNA reveals interactions between PRORP and the pre-tRNA extremities (5′-leader and 3′-trailer), and between PRORP and TRMT10C. We show that the first two helices of TRMT10C interact with the PPR domain of PRORP, clamping PRORP on the pre-tRNA. PRORP is not active on the pre-tRNA pair when these interactions are abolished. Our results thus favor a model where the TRMT10C/SDR5C1 complex helps position or maintain PRORP on the pre-tRNA, in agreement with a recent kinetic study[32].

The TRMT10C/SDR5C1 complex recognises the entire mitochondrial pre-tRNA structure, acting as a checkpoint for the integrity of each pre-tRNA stem. For instance, tRNA$^{Ser}$(AGY), that lacks the D-stem, is not bound by the TRMT10C/SDR5C1 complex[40]. The inability of TRMT10C/SDR5C1 to bind a pre-tRNA prevents PRORP from processing its 5′-leader, thereby blocking ELAC2 from cleaving the 3′-trailer, in turn blocking TRNT1 from adding the 3′-CCA sequence. This blocking of pre-tRNA maturation, when tRNAs cannot bind to TRMT10C/SDR5C1, gives the subcomplex a role in tRNA quality-control and could prevent pre-tRNAs for which an error occurs during transcription from entering the maturation process. tRNA$^{Ser}$(AGY) is found in a tRNA triplet, where it is bordered by more canonical pre-tRNAs upstream and downstream. The processing of the bordering pre-tRNA$^{His}$ by ELAC2 simultaneously releases 3′-processed tRNA$^{His}$ and 5′-processed tRNA$^{Ser}$(AGY)[23]. As such, no processing is specifically required for the structurally-fragile mitochondrial pre-tRNA that would not otherwise be recognised by TRMT10C/SDR5C1. Our structure of ELAC2/mt-pre-tRNA$^{His(0,Ser)}$ shows in detail this specific situation in which pre-tRNA$^{Ser}$(AGY) is released by 3′-end processing of the bordering TRMT10C-bound mt-pre-tRNA$^{His}$, providing a structural interpretation of the results from previous studies on the processing of this pair[23]. A second cluster of tRNAs is encoded on the mt-DNA light-strand (tRNA$^{Tyr}$-tRNA$^{Cys}$-tRNA$^{Asn}$-tRNA$^{Ala}$). TRMT10C/SDR5C1 likely binds to each tRNA and further studies will be needed to determine the chronology of processing events and which enzyme, either PRORP or ELAC2, cleaves which junction.

Metazoan mitochondrial tRNAs present low structural stability and conserved elements, such the elbow formed by the interaction of the tRNA D- and T-loops, are variable. While other RNase P, RNase Z and 3′-nucleotidyltransferase enzymes interact with the conserved tRNA elbow[43–46], the mitochondrial enzymes interact with the T-loop of pre-tRNAs and with the TRMT10C NTD. The interaction with the TRMT10C NTD is required for PRORP activity, but not for ELAC2 and TRNT1 (this study and[27]) that are both active on the pre-tRNA alone and on the TRMT10C/SDR5C1-bound pre-tRNA. The question that remains is what will be the status of the pre-tRNAs, whether bound to TRMT10C/SDR5C1 or not, after PRORP processing. So far, the question of what triggers the dissociation between the TRMT10C/SDR5C1 complex and tRNA remains unanswered, the $N^1$-methylation not being responsible for tRNA release[27,32]. In vitro, multiple turnovers could not be observed with TRMT10C[27,32], and the pre-tRNA was shown to remain bound to the TRMT10C/SDR5C1 complex[27]. Our structures show that the TRMT10C/SDR5C1 complex can be a scaffold that presents mitochondrial pre-tRNAs as substrates for chemical modification,

i.e. methylation, and nucleolytic cleavages at defined 5′ and 3′ sites and, finally 3′-CCA addition (Fig. 1). For the remaining steps of tRNA maturation, such as the incorporation of nucleotide chemical modifications, the TRMT10C/SDR5C1 complex is not required. This leads us to suggest a model for mitochondrial RNA processing and tRNA maturation, which is summarised in a movie (Supplementary Movie 1). First, TRMT10C and SDR5C1 form a subcomplex, which can bind and methylate the pre-tRNA. The SDR5C1 tetramer acts as a platform that can bind two TRMT10C/pre-tRNA units, one on each face of the tetramer (Supplementary Fig. 6), suggesting that two pre-tRNAs could be processed simultaneously. PRORP is either already bound to the pre-tRNA or recruited by the TRMT10C/SDR5C1/pre-tRNA subcomplex to catalyze 5′-cleavage. The next steps in pre-tRNA processing, i.e. 3′-cleavage by ELAC2 and 3′-CCA addition, can occur through a factor exchange between PRORP and ELAC2, and then between ELAC2 and TRNT1. The mitochondrial tRNA methyltransferase subcomplex can thus be viewed as a factor that compensates for the structural variability in the mitochondrial tRNA substrates and provides an additional attachment point for the maturation enzymes to guarantee proper maturation of structurally-degenerate pre-tRNAs.

Genetic mitochondrial disorders originate from mutations in mt-DNA or in nuclear genes encoding mitochondrial proteins. Most of the disease-causing mt-DNA point mutations are mapped to mitochondrial tRNAs (MITOMAP, www.mitomap.org)[47] and are associated with devastating neurological disorders, cardiovascular myopathies and hypertension (reviewed in[48]). These point mutations can change the structure of mitochondrial pre-tRNAs, their processing, their aminoacylation and their modification contents[15,48,49]. pre-tRNAs, with disease mutations located near the elbow region, were shown to still bind to RNase P but were proposed not to be correctly oriented in the active site of TRMT10C and PRORP for optimal methyltransferase and cleavage activity[49]. The interaction between the PRORP/TRMT10C and the pre-tRNA, requires to position correctly the active site of PRORP, could explain these defects. Our structural data also provides information on how specific mutations in conserved residues of mitochondrial tRNA maturation enzymes can lead to loss of function associated with human mitochondrial diseases (Supplementary Fig. 14). Most of these mutations cause defects in mitochondrial tRNA maturation and respiratory chain deficiency in patients[49–56]. For TRMT10C, the disease-associated mutations, R181 and L182, are involved in the stabilisation of the distinctive anticodon loop conformation observed in all structures of tRNA bound to the TRMT10C/SDR5C1 complex, while T272 is located in a large loop and makes hydrophobic contact with Y244, a conserved residue of TRMT10C. This interaction is likely important to maintain the folding of the MTase domain of TRMT10C. For PRORP, the disease-associated mutations are located only in the catalytic domain (A485, A434, R521C) and close to the $Mg^{2+}$ active site (A412 and R445). For ELAC2, the disease-associated mutations are located either close to the active site of ELAC2 (T520, V642, K660, C670, A680, Y729) or at the junction between the two β-lactamase domains of ELAC2 (R68, I153, F154, S347, Q388, L422, L423, P493, R564, H749). The first group of residues stabilises the structure of the active site of ELAC2 and the second group maintain the interaction between the two β-lactamase domains. This is crucial for enzyme activity since one β-lactamase domain is used for catalysis and the other for tRNA binding. Other mutations, located in the Nter and Cter β-lacD (Y75, G132 and R781) are involved in tRNA binding, like G132 and R781 that bind the acceptor arm of tRNA^His. For TRNT1, most of the disease-linked mutations are close to the active site (R99, T110, D128, A148, T154, I155, M158, D163, L166, Y173, R190, R203, H215), and other mutations (I223, I326, K416, S418) are located at the junction between two different domains of TRNT1, potentially affecting the global architecture of TRNT1.

The structures of ELAC2 and TRNT1 in complex with mitochondrial pre-tRNA also provide insights into their recognition of cytoplasmic tRNAs. While they interact with the T-loop of mitochondrial tRNAs to drive the tRNA binding and position their active site at the extremity of the acceptor stem, ELAC2 and TRNT1 can interact using the same organisation with the elbow of cytoplasmic tRNAs formed by the interaction of the D- and T-loops. Indeed, aligning pre-tRNA^His with a cytosolic human tRNA, like tRNA^Lys_3 (PDB 1FIR), provides a model of ELAC2/tRNA where the acceptor- and T-arms of tRNA^Lys_3 are wedged between the large helix α24 (residues 772-794) of the Cter β-lactD, at the level of the acceptor arm of tRNA^Lys_3, and the domain that protrudes from the Nter β-lactD, at the level of the tRNA elbow (Fig. 7a). This recognition mode resembles that used by bacterial dimeric short RNase Z to bind tRNA[57]. Similarly, aligning mitochondrial tRNA^Ile with the human cytosolic tRNA^Lys_3 suggests that TRNT1 interacts with the elbow of cytosolic tRNA at the level of the G19-C56 base pair (Fig. 7b). Like ELAC2, TRNT1 likely interacts with the elbow of cytosolic tRNAs and with the T-loop of mitochondrial tRNAs since the elbow is less structured or stable due to the difference in length of the D- and T-arms in mitochondrial tRNAs compared to cytosolic tRNAs. tRNA^Lys_3, which has a stable elbow, can also be placed in the complex of mitochondrial RNase P, Z and 3′-CCA addition described in this paper (Supplementary Fig. 13). This shows how more canonical mitochondrial pre-tRNAs, with more stable elbows, can be bound by TRMT10C/SDR5C1 complex for 5′- and 3′-processing and 3′-CCA addition.

Finally, this study paves the way for future research into the link between mitochondrial tRNA maturation and diseases, into the mechanisms of the other stages of mitochondrial tRNA maturation and into the post-transcriptional regulation of mitochondrial gene expression.

## Methods

### Expression and purification of TRMT10C/2, PRORP, ELAC2 and TRNT1

Human TRMT10C (UniProt Q7L0Y3, residues 40-403) and SDR5C1 (UniProt Q9971, residues 1-261) were co-expressed as previously described[26]. TRMT10C/SDR5C1-expressing plasmid was transformed into *E. coli* BL21(DE3) Rosetta competent cells, cultured in Lysogenic Broth (LB) at 37 °C. Protein expression was induced with 0.1 mM isopropyl β-D-1-thiogalactopyranoside (IPTG) overnight at 18 °C. The cell pellet from 3 L culture of TRMT10C/SDR5C1 was lysed by sonication in a buffer containing 50 mM Tris pH 8.0, 1 M NaCl, 5% glycerol (v/v), 1 mM TCEP, 1 mM PMSF, 3 mM EDTA and clarified by centrifugation. The clarified lysate was applied to a 5 mL HisTrap HP column (Cytiva) and eluted with a gradient of 20-250 mM imidazole. TRMT10C/SDR5C1 was then purified by size exclusion chromatography (SEC) on a HiLoad 26/600 Superdex 200 pg (Cytiva) column and by affinity chromatography on a 5 mL HiTrap Heparin HP column (Cytiva). The TRMT10C/SDR5C1 sample was then fractionated by size exclusion chromatography (SEC) using a HiLoad 26/600 Superdex 200 pg (Cytiva). The enriched fractions of TRMT10C/SDR5C1 were concentrated using an Amicon® 10,000 MWCO (Merck Millipore) and stored at -80 °C in the SEC buffer (50 mM Tris pH 8.0, 100 mM NaCl, 1 mM TCEP).

Human PRORP (UniProt O15091, residues 45-583), human ELAC2 (UniProt Q9BQ52, residues 31-826) and TRNT1 (UniProt Q96Q11, residues 42-434) were expressed from plasmids constructed from a pET28a vector modified to incorporate an N-terminal His6-tag followed by a Tobacco Etch Virus protease (TEV) cleavage site and from a synthetic gene with optimized codons for the expression in *E. coli*. The PRORP-expressing plasmid was transformed into *E. coli* BL21(DE3) Rosetta competent cells, cultured in LB at 37 °C, and protein expression was induced with 0.1 mM IPTG overnight at 18 °C. Cell pellet from 2 L culture of PRORP was lysed by sonication in a buffer containing 50 mM Tris pH 8.0, 1 M NaCl, 5% glycerol (v/v), 1 mM TCEP, 1 mM PMSF, 3 mM EDTA and clarified by centrifugation. The clarified lysate was applied to a 5 mL HisTrap HP column (Cytiva) and eluted with a gradient of 20-250 mM imidazole. PRORP was treated with the TEV

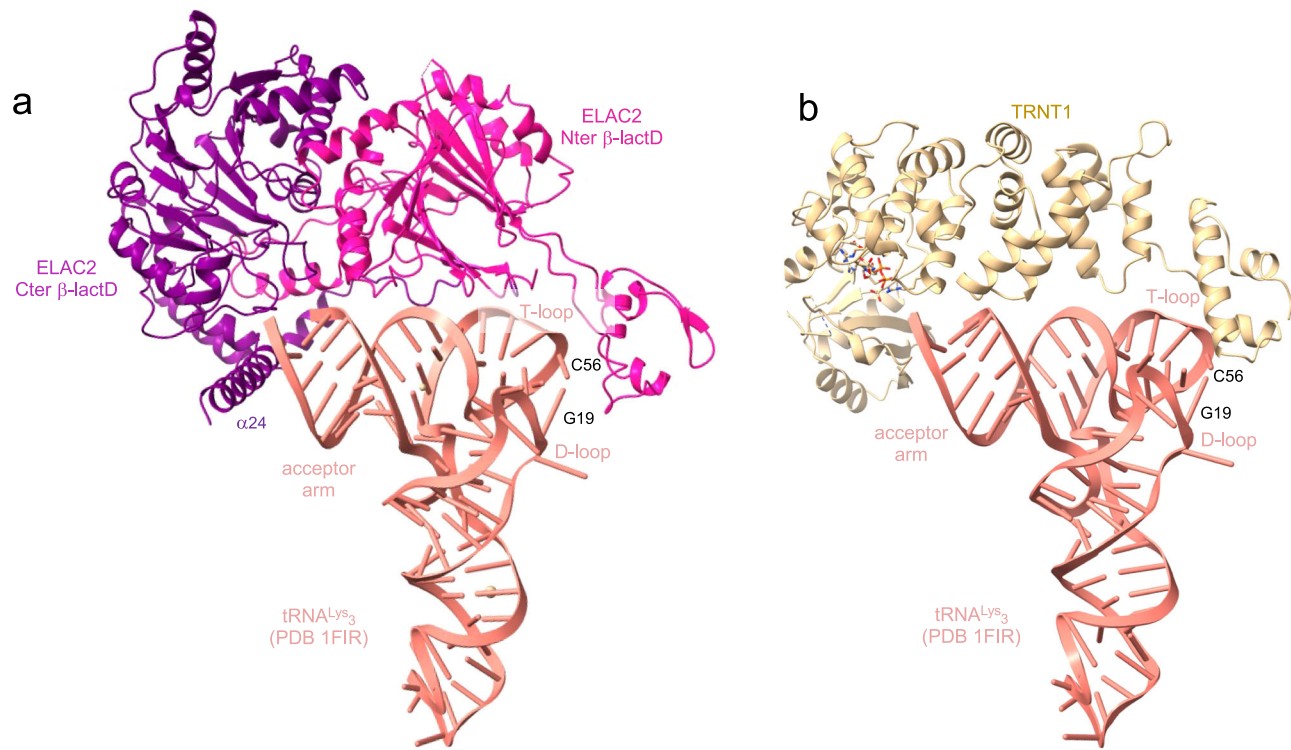

**Fig. 7 | ELAC2 and TRNT1 interactions with human cytosolic tRNAs. a** Model of the ELAC2/human cytosolic tRNA$^{Lys}_3$ complex. **b** Model of the TRNT1/human cytosolic tRNA$^{Lys}_3$ complex.

protease for 12 h at 4 °C. Further purification of PRORP was performed by SEC on a HiLoad 26/600 Superdex 75 pg (Cytiva). PRORP was then concentrated in the SEC buffer using an Amicon® 10,000 MWCO (Merck Millipore) and stored at -80 °C.

The ELAC2-expressing plasmid was transformed into *E. coli* KRX competent cells (Promega), cultured in LB at 37 °C and protein expression was induced with 0.1 mM IPTG and 0.05% rhamnose overnight at 18 °C. Cell pellet from 6 L culture of ELAC2 was lysed by sonication in a buffer containing 50 mM Tris pH 8.0, 1 M NaCl, 5% glycerol (v/v), 1 mM TCEP, 1 mM PMSF, 3 mM EDTA and clarified by centrifugation. The clarified lysate was applied to a 5 mL HisTrap HP column (Cytiva) and eluted with a gradient of 20-500 mM imidazole. ELAC2 was applied to a HiPrep 26/10 Desalting (Cytiva) and to a 6 mL Resource Q column (Cytiva) and eluted with a gradient of 50-1000 mM NaCl. ELAC2 was treated with the TEV protease for 12 h at 4 °C. Further purification of ELAC2 was performed by SEC on a HiLoad 26/600 Superdex 75 pg (Cytiva), ELAC2 was then concentrated using an Amicon® 50,000 MWCO (Merck Millipore) and stored at -80 °C in the SEC buffer.

The TRNT1-expression plasmid was transformed into E. coli KRX competent cells (Promega), cultured in LB at 37 °C and protein expression was induced with 0.1 mM IPTG and 0.05% rhamnose overnight at 18 °C. Cell pellet from 3 L culture of TRNT1 was lysed by sonication in a buffer containing 20 mM HEPES pH 7.5, 1 M NaCl, 5% glycerol (v/v), 1 mM TCEP, 1 mM PMSF, 3 mM EDTA and clarified by centrifugation. The clarified lysate was applied to a 5 mL HisTrap HP column (Cytiva) and eluted with a gradient of 20-500 mM imidazole. TRNT1 was then applied to a HiPrep 26/10 Desalting (Cytiva) and to a 5 mL HiTrap Heparin HP column (Cytiva) and eluted with a gradient of 50-1000 mM NaCl. TRNT1 was treated with the TEV protease for 12 h at 4 °C. Further purification of TRNT1 was performed by SEC on a HiLoad 26/600 Superdex 75 pg column (Cytiva). TRNT1 was concentrated using an Amicon® 10,000 MWCO (Merck Millipore) and store at -80 °C in the SEC buffer.

All mutations were generated using the QuikChange site-directed mutagenesis kit (Agilent) and the presence of the DNA changes was verified by sequencing. All mutants were expressed and purified like WT proteins.

## Preparation of pre-tRNA$^{His-Ser}$, pre-tRNA$^{His(0,Ser)}$, pre-tRNA$^{Ile}$, pre-tRNA$^{Ile(0,4)}$ and tRNA$^{Ile}$

Human mt pre-tRNA$^{His-Ser}$ with 5 or 0 nucleotides (nt) and 4 nt as 5′-leader and 3′-trailer sequences, respectively, pre-tRNA$^{Ile}$ with 5 nt and 4 nt as 5′-leader and 3′-trailer sequences, respectively, and tRNA$^{Ile}$ with no 5′-leader and 3′-trailer were produced by in vitro transcription using T7 RNA polymerase. The sequences of the DNA templates were: 5′TGAGAAAGCCATGTTGTTAGACATGGGGGCATGAGTTAGCAGTTCT TGTGAGCTTTCTCGGTAAATAAGGGGTCGTAAGCCTCTGTTGTCAGA TTCACAATCTGATGTTTTGGTTAAACTATATTTACAAGCC" for the pre-tRNA$^{His-Ser}$ with 5 nt for the 5′-leader and tRNA$^{Ser}$ for 3′-trailer (Supplementary Fig. 1a); 5′TGAGAAAGCCATGTTGTTAGACATGGG GGCATGAGTTAGCAGTTCTTGTGAGCTTTCTCGGTAAATAAGGGGTC GTAAGCCTCTGTTGTCAGATTCACAATCTGATGTTTTGGTTAAACTAT ATTTAC3′ for the pre-tRNA$^{His(0,Ser)}$ with 0 nt for the 5′-leader and tRNA$^{Ser}$ for the 3′-trailer (Supplementary Fig. 1a); 5′GTCCTAGAAATAA GGGGGTTTAAGCTCCTATTATTTACTCTATCAAAGTAACTCTTTTATC AGACATATTTCTTACCC3′ for pre-tRNA$^{Ile}$ with 5 nt for the 5′-leader and 4 nt for the 3′-trailer sequences (Supplementary Fig. 1a); 5′GTCC TGGAAATAAGGGGGTTTAAGCTCCTATTATTTACTCTATCAAAGTAAC TCTTTTATCAGACATATTTCC3′ for pre-tRNA$^{Ile(0,4)}$ with no nt for the 5′-leader and 4 nt for the 3′-trailer (Supplementary Fig. 1a) and 5′TGGA AATAAGGGGGTTTAAGCTCCTATTATTTACTCTATCAAAGTAACTCTT TTATCAGACATATTTCC3′ for tRNA$^{Ile}$ with no 5′-leader and no 3′-trailer (Supplementary Fig. 1a).

T7 RNA polymerase prefers a G nucleotide to start the transcription, the two last constructs were thus designed to start with a GC base pair (Supplementary Fig. 1a). The T7 promotor sequence 5′TATAGT-GAGTCGTATTA3′ was added at the 3′ of these DNA sequences to

promote the transcription of the tRNA by the T7 RNA polymerase. DNA templates for pre-tRNA$^{His-Ser}$ were obtained using two consecutive PCR, as previously described[58] whereas DNA templates for pre-tRNA$^{Ile}$ and tRNA$^{Ile}$ and T7 promotor primer were purchased from Eurogentec. RNA transcripts were purified by ion exchange chromatography (MonoQ™ 10/100 GL, Cytiva) under native conditions, i.e. 20 mM Na-phosphate pH 6.5 with a NaCl gradient from 50 mM to 1 M, dialysed extensively against 1 mM Na-phosphate pH 6.5, then refolded by heating at 95 °C for 10 min and cooling down slowly at room temperature. Purified transcripts were finally placed in a buffer of 10 mM Na-phosphate pH 6.5, 2 mM MgCl₂ and concentrated using Amicon® 3000 MWCO (Merck Millipore).

### Methyltransferase activity assays

A reaction contained 10 pmol of pre-tRNA supplemented with 5 pmol S-adenosyl-L-[methyl-$^3$H]-methionine (80 Ci/mmol; American Radiolabeled Chemicals, Inc.) and followed by the addition of 6 pmol wild-type or mutant TRMT10C/SDR5C1 and/or buffer alone. All reactions were carried out at 37 °C for 1 h in a 15 µL final volume in a buffer of 50 mM Tris pH 8.5, 20 mM NaCl, 5 mM MgCl₂, 1 mM DTT), then quenched in 1.5 mL of ice cold 5% (w/v) trichloroacetic acid and incubated on ice for at least 10 min. The precipitates were collected by filtration using GF/C filters (Whatman) under vacuum and washed four times with 5 mL of cold 5% (w/v) trichloroacetic acid. The washed and dried filters, using 5 ml of EtOH, were then placed in vials containing 5 mL of liquid scintillation cocktail (PerkinElmer), shaken for 15 min and counted in an Hidex 300 SL scintillation counter (LabLogic ScienceTec). Standard deviations are calculated from three technical replicates. Data were analysed by one-way ANOVA using a F test statistic.

### RNase P and RNase Z cleavage activity assays

All assays were carried out in 60-µl reactions in a cleavage buffer containing 20 mM Na-HEPES pH 7.5, 130 mM KCl, 2 mM MgCl₂, 2 mM TCEP, 20 µM NADH, 5 µM SAM, 1 u/µl of ribonuclease inhibitors (RNasin® from Promega). The reaction mixes included 200 nM pre-tRNA, 1 µM TRMT10C/SDR5C1 complex (wild-type or deletion mutants), 50 nM PRORP (wild-type or D479A mutant) and/or 50 nM ELAC2 (wild-type, H548A mutant or deletion mutants). Reactions were incubated at 30 °C or 37 °C, aliquots were collected at these set times: 5, 10, 15, 30 and 60 minutes. Reactions were quenched with an addition of an equal volume of 2X RNA loading dye buffer (8 M urea, 5 mM Tris pH 7.5, 20 mM EDTA pH 8.0, 0.5% xylene cyanol), then 0.8 units of proteinase K from *Tritirachium album* (New England Biolabs Inc.) were added. The quenched reactions were incubated at 30 °C for 60 min. The reaction products were heated 5 min at 95 °C and analysed by denaturing PAGE on a 10 % or 16% polyacrylamide (acrylamide:bis-acrylamide ratio of 19:1) Tris-Borate-EDTA urea gels, for the pre-tRNA$^{His-Ser}$ and pre-tRNA$^{Ile}$, respectively. Gels were stained for 20 min with SYBR™ Green II (Invitrogen™), imaged using a Bio-Rad ChemiDoc imaging system and quantified using ImageJ[59].

### Assays of CCA-adding activity

Assays were carried out in 90-µl reactions in a buffer containing 20 mM Na-HEPES pH 7.5, 150 mM KCl, 5 mM MgCl₂, 2 mM TCEP, 0.5 mM cytidine-5'-triphosphate (CTP) and 0.5 mM adenosine-5'-triphosphate (ATP). The reaction mixes included 200 nM tRNA$^{Ile}$, 800 nM TRMT10C/SDR5C1 complex and 50 nM TRNT1. Reactions were incubated at 30 °C and aliquots were collected at 15 and 30 minutes. Reactions were quenched with the addition of an equal volume of 2X RNA loading dye buffer and 0.8 units of proteinase K from *Tritirachium album* (New England Biolabs Inc.) were added. The quenched reactions were incubated at 30 °C for 60 min. The reaction products were heated 5 min at 95 °C and analysed by denaturing PAGE on a 14 % polyacrylamide (acrylamide:bis-acrylamide ratio of 19:1) Tris-Borate-EDTA urea gels. Gels were stained for 10 min with ethidium bromide (BET) and imaged using a Bio-Rad ChemiDoc imaging system.

### Cryo-EM sample preparation, data collection and image processing

Samples for cryo-EM analysis were prepared in a buffer of 50 mM Tris pH 8.0, 100 mM NaCl and 1 mM TCEP by mixing TRMT10C/SDR5C1 with SAH and NADH, then the pre-tRNA and last one maturation enzyme (PRORP, ELAC2 or TRNT1). For the complex with TRNT1, the buffer was 25 mM Tris pH 8.0, 50 mM NaCl, 0.5 mM TCEP and 0.2 mM MgCl₂. For complex 1 (TRMT10C/SDR5C1/SAH/NADH/pre-tRNA$^{Ile}$) formation, 5 µM TRMT10C/SDR5C1, 50 µM SAH, 50 µM NADH and 25 µM pre-tRNA$^{Ile}$ were mixed. For complex 2 (TRMT10C/SDR5C1/SAH/NADH/pre-tRNA$^{His-Ser}$/PRORP$_{D479A}$), a mixture of 4 µM of TRMT10C/SDR5C1, 40 µM SAH, 40 µM NADH, 20 µM pre-tRNA$^{His-Ser}$ and 10 µM PRORP$_{D479A}$ was used to make the grids. For complex 3 (TRMT10C/SDR5C1/SAH/NADH/pre-tRNA$^{His(0,Ser)}$/ELAC2$_{H548A}$), 1 µM TRMT10C/SDR5C1, 10 µM SAH, 10 µM NADH, 5 µM pre-tRNA$^{His(0,Ser)}$ and 5 µM ELAC2$_{H548A}$ were mixed to make the grids. For complex 4 (TRMT10C/SDR5C1/SAH/NADH/tRNA$^{Ile}$/TRNT1/CDP) formation, 1.5 µM TRMT10C/SDR5C1, 15 µM SAH, 15 µM NADH, 1 µM tRNA$^{Ile}$, 10 µM TRNT1, 0.5 mM CDP were mixed.

Quantifoil Cu 300 1.2/1.3 grids were treated by glow-discharge using a PELCO easiGLOW device with settings of 60 s, 25 mA, 0.39 mBar. For vitrification, 4 µL of sample was applied to the grids and incubated for 30 s. The grids were blotted with a blot force of -5 for 3 s for complex 1, 2 and 4 and a blot force of 0 for 3 s for complex 3, using a Vitrobot Mark IV (ThermoFisher) at 4 °C and 100% humidity immediately before plunge-freezing in liquid ethane. Grids were subsequently stored in liquid nitrogen until screening and collection. Datasets were collected on a 300 kV Titan Krios equipped with a Gatan K3 detector. Data collection parameters are summarized in Supplementary Table 1. Particles were picked and pre-processed (motion correction, CTF estimation and particle extraction) with WARP[60]. After migration of particle stacks to cryoSPARC (v3.1 and V4.0)[61] 2D classification, ab initio reconstruction and 3D non-uniform refinement yielded the final reconstructions[62]. The dataset of sample TRMT10C/SDR5C1/SAH/NADH/pre-tRNA$^{Ile}$ was processed in Relion 3.1[63] (Supplementary Fig. 3). Micrographs were motion-corrected with MotionCor2 and CTFs were estimated using CTFFIND-4.1. 2D class averages were generated from an initial set of particles that were picked with the Laplacian-of-Gaussian autopicking algorithm in Relion 3.1 and a selection was used for template-based auto-picking in Relion. Iterative rounds of 2D- and 3D-classification yielded a 3.0 Å resolution reconstruction. To improve the resolution of subassemblies, local refinements with soft masks around the area of interest (Supplementary Fig. 3-6) were performed. All maps were anisotropically sharpened as implemented within Phenix[64]. The locally refined and sharpened subassembly maps were combined to make the composite maps.

To improve the resolution of subassemblies, local refinements with soft masks around the area of interest (Supplementary Fig. 3-6) were performed.

### Model building and refinement

The cryo-EM structure of TRMT10C/SDR5C1/PRORP/pre-tRNA$^{Tyr}$ (PDB 7ONU)[25] was fitted into the density map of the human mt RNase P in complex with pre-tRNA$^{His-Ser}$. The structure of the MTase domain of TRMT10C in complex with SAH (PDB 5NFJ)[26] was used to build the MTase domain of TRMT10C bound to SAH. The AlphaFold[38,39] model of TRMT10C (AF-Q7L0Y3) was used to construct three parts of TRMT10C NTD that are not present in 7ONU, residues 61-91 that interact with PRORP, residues 157-174 at the end of the NTD and residues 386-403 of the MTase domain. In all complexes, the pre-tRNA 5'-leader and 3'-trailer were built de novo, and the AlphaFold[38,39] models of PRORP (AF-O15091), ELAC2 (AF-Q9BQ52) and TRNT1 (AF-Q96Q11) were used to

build the maturation enzymes. The TRMT10C/SDR5C1/pre-tRNA[His] sub-complex of complex 2 was fitted in the density maps of complex 1, 3 and 4 to start the building of models for these complexes. Models were docked to the density in Chimera-1.11.2[65] and ChimeraX V1.5[66], and 5′-leaders and 3′-trailers were modelled manually in Coot-0-8-9-1[67,68] from the CCP4 suite of programs[69]. Model of complex 4 was used to build the tRNA[Ile] model in complex 1 where the quality of the density map for tRNA[Ile] is lower. After model building, structures were subjected to geometrical optimisation with ISOLDE[70] and cycles of real-space refinement in Phenix[64] against the locally filtered cryo-EM maps and manual fitting in Coot. Model vs. map statistics are reported in Supplementary Table 1.

### Differential scanning fluorimetry (DSF)

DSF was performed in a 96-well plate using a CFX96 Touch real-time PCR detection system (Bio-Rad) with excitation and emission filters of 450-490 and 515-530 nm, respectively. Each well contained 2 μL protein in a buffer of 20 mM HEPES pH 7.5, 150 mM NaCl to a final concentration of 5 μM, 2 μL of SYPRO Orange diluted 5000-fold in buffer from the manufacturer's stock (Invitrogen). Fluorescence intensities were measured from 25 to 95 °C with a ramp rate of 1 °C/min. Tm-values were determined by curve-fitting using GraphPad Prism v.5.01 software of three technical replicates as previously described[71]. Standard deviations are calculated from three technical replicates. Data were analysed by one-way ANOVA using a Dunnett's multiple comparison test.

### Quantification and statistical analysis

Quantification methods and tools are described in each relevant section of the methods and in figure legends.

### Reporting summary

Further information on research design is available in the Nature Portfolio Reporting Summary linked to this article.

## Data availability

The data supporting the findings of this study are available from the corresponding authors upon request. The atomic coordinates and the cryo-EM maps have been deposited in the Protein data Bank (PDB) and in the Electron Microscopy Data Bank (EMDB), respectively, under the following accession codes: PDB 8CBO and EMD-16547 for Complex 1, PDB 8CBK and EMD-16544 for Complex 2, PDB ID 8CBL and EMD-16543 for Complex 3, PDB 8CBM and EMD-16545 for Complex 4. Source data are provided with this paper.

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

## Acknowledgements

We thank Pedro Guiomar and Ciarán Condon for careful reading of the manuscript. This work was supported by funds from the CNRS and the University Paris Cité (UMR8261), the Agence Nationale de la Recherche (Project ARNTools ANR-19-CE07-0028 and the Labex Dynamo ANR-11-LABX- 0011). J. J. R. and B. F. L. are supported by Wellcome Trust Investigator Awards (200873/Z/16/Z and 222451/Z/21/Z). We acknowledge access to Titan Krios at Diamond eBIC lightsource.

## Author contributions

C.T. conceived, designed the experiments and supervised the project. V.M. prepared the protein samples (WT and mutants), the complexes for the cryo-EM analysis and performed the methyltransferase assays. M.C. prepared the tRNA samples for the cryo-EM analysis and performed the cleavage activity assays with input from P.B. S.O. and W.Y. provided the expression plasmid of TRMT10C/SDR5C1 and protocol to purify the complex. B.F.L. prepared the grids. S.W.H., J.J.R., D.Y.C. and B.F.L. performed cryo-EM data collection and processing. V.M. and C.T. performed the building of models and refinement. V.M., B.F.L. and C.T. analysed the data and C.T. prepared the manuscript with input from all authors. All authors discussed the data analysis, critically reviewed the manuscript, and approved the final version.

## Competing interests

The authors declare no competing interests.
