## [Peer Review File · Nature Communications]

Structural basis for human mitochondrial tRNA maturationREVIEWER COMMENTS

Reviewer #1 (Remarks to the Author):

In mammalian mtDNA, tRNA genes are encoded between rRNA and mRNA genes without any intervening sequence. Once tRNAs are processed by RNases P and ELAC2 from the long precursor RNAs transcribed from mtDNA, rRNAs and mRNAs are generated. Then, the tRNAs undergo 3'-CCA addition by TRNT1. During tRNA processing, a wide variety of post-transcriptional modifications are introduced. Unique to metazoan mitochondria, RNase P is an RNA free multi-enzyme complex composed of PRORP, TRMT10C and SDR5C1. In this study, authors solved several cryoEM structures of precursor tRNAs bound by these maturation enzymes. The complex of pre-tRNAs bound by TRMT10C/SDR5C1 and PRORPP are already reported by another group a few years ago. So, the originality of this study is two complex structures of pre-tRNA-TRMT10C/SDR5C1 together with ELAC2 or TRNT1. This study surely contributes to our deep understanding of tRNA maturation process in human mitochondria. This reviewer has several comments and concerns that need to be addressed before publication.

Since PRORP actually recognizes tRNA and TRMT10C, 5' processing of tRNA should coordinate with m9R methylation. However, this reviewer does not see any clear interaction between ELACS and TRMT10C, and TRNT1 and TRMT10C, indicating that 3' processing and CCA addition take place independent of m9R methylation. Because various tRNA-modifying enzymes are involved in tRNA maturation in mitochondria, it is quite natural to think other tRNA-modifying enzymes also recognize tRNA precursors instead of TRMT10C/SDR5C1 binding when their 3' ends are processed. Authors should mention and properly discuss other pathways for tRNA maturation.

In terms of cryoEM maps, the lower parts of complexes (SDR5C1, TRMT10C, tRNA anticodon loop) show high resolution. In contrast, the upper part (tRNA Ser, PROPP, ELAC2, TRNT1) show relatively low resolution (over 4Å). Authors cannot discuss anything about the side chain position of the specific residues throughout the manuscript.

Because these maps are 2.7-3.2Å resolution, this reviewer supposes H₂O should not be modelled. H₂O can be modelled if resolution is lower than 2-2.2Å resolution. In addition, this reviewer supposes several Mg ions exist around tRNA (for example, near A9 in sample2), it's better to model these Mg ions first.

Local resolution maps of sample2-4 maps (as seen in Sup Fig1E) are required, because overall resolution of refined maps (showed in Sup Fig345D) is not consistent with local resolution of each region.

Page 2 (3rd paragraph): N1-methylation. "N" should be italic and "1" should be superscript.

Page 3 (2nd paragraph): "significantly" should be "significant"

Page 5, lines 1-4: It's useful for readers to describe the constructs of recombinant proteins whether they have MTS or not.

Page 5, line 5: Appropriate references are necessary for this sentence, "

Page 5, line 7: An explanation should be included for notation of His(0, Ser) and Ile(0,4) in the pre-tRNA^{His}(0, Ser) and pre-tRNA^{Ile}(0, 4) respectively.

Page 7: It is better to clearly mention the region of ELAC2 that clashes with pre-tRNA 5'-leader in the main text. In addition, the reviewer would like the authors to check if the region is conserved in other organisms, aiming to validate the generality of their findings.

Page 8 Line8: "G1 and U2 of tRNAHis..." should be "G1 and U-1 of tRNAHis..."

Page 8, line 10: Explain "310 helix".

Page 8, the second paragraph: The authors should use figures to illustrate what is written here.

Page 8: Regarding the results in Fig 3f: The delta(1-80) mutant appears to be active, and the authors should explain why. The authors should also comment in the main text on the results of the delta(1-166) mutant.

Page 10 line 2: Fig. 4b should be 4c?

Page 10 (1st paragraph): no Figure for surface charge in Fig. 4a?

Page 10 Last paragraph: Replacing of pre-tRNA with cytoplasmic tRNA from other model is not impressive, because ELAC2 structure would be changed by tRNA substrate. Especially, the position of protrusion in the Nter β -lactD would be easily changed. This reviewer does not understand what the authors intend to describe with this model in Figure 5g. It should be removed or relocated to SI figure.

Page 12 Line19: ATP accommodation model should be shown in the figure.

Page 13: "The conserved residue L180 interacts with G34 that is further stabilized by chain C of SDR5C1 via residues (98-105) from a lysine-rich loop that also interacts with U32." But, SDR5C1 is not shown in Fig 6.

Page 15: "For the human TRMT10A and the *S. cerevisiae* Trm10 enzymes..." Provide the AlphaFold models of these enzymes (AF-Q12400 and AF-Q8TBZ6) in SI figure.

Page 16: Based on SAH model, SAM model can be generated to explain the reaction mechanism mediated by some specific residues.

Page 17, last sentence: Many pathogenic mutations have been reported in human mitochondrial tRNA. This reviewer asks the authors to mention whether any of the pre-tRNA bases involved in the interactions with each protein identified in this paper are related to pathogenic mutations. Also, it is better to mention whether pathogenic mutations have been reported in the proteins as well.

Page 18, lines 3-5: The authors should explain exactly what kind of transcripts they are referring to here as non-canonical mitochondrial pre-tRNA.

Page 18: Besides His-Ser pair, it is interesting for this reviewer to discuss whether this character is preserved or not in the other mt-tRNA clusters?

Fig.1c: Pre-tRNAHis-Ser should be Pre-tRNAHis(0,Ser)

Fig. 2b: the presence of 5'-reader prevents cleavage by ELAC2. The cleavage activity of ELAC2 for each tRNA should be quantified by the band intensity of the product.

Fig.3c: A479 should be D479A

Fig. 3f: The line with "Pre-tRNAHis(0,Ser)" is overlapped with the gel image (Ref Lane). Please improve.

Fig. 5c: The density of CDP is not clearly visualized. It's better to show this figure with different angle. If the density of phosphate group is observed, discuss binding residues with phosphate group.

Sample 2 and 3.

It is confusing that model numbering and Figure legend are different at tRNAs. Mg1 and Zn1 densities are not shown in there maps. Please move this part to discussion section.

Sample4.

The densities of CDP and surrounding residues of TRNT1 are weak, and binding mode against CDP is not clearly defined. You should move this part in discussion section.

Reviewer #2 (Remarks to the Author):

In this manuscript, V. Meynier and co-workers. have combined structural and functional approaches to give a comprehensive description of the machinery required for the maturation of human mitochondrial tRNA. In this organelle, precursors to mt-tRNA are matured from polycistronic transcripts by a series of events, namely endonucleolytic cleavages, addition of a trinucleotide moiety at their 3' end (3'-CCA), as well as chemical modifications such as methylations. These maturation events are performed through the association of pre-mt-tRNA to a maturation platform composed of proteins TRMT10C and SDR5C1, with which the enzymes PRORP, ELAC and TRNT1 will associate for 5'-, 3'-endonucleolytic cleavage and 3'-CCA addition, respectively. Although structures of mt-RnaseP (TRMT10C/SDR5C1/PROP) complex bound to pre-tRNA^{Tyr} have recently been published, little is known regarding how the rest of the tRNA maturation sequence happens. The authors have thus used cryo-electron microscopy to propose four atomic 3D structures of maturing pre-mt-tRNA bound to the TRMT10C/SDR5C1 platform alone or associated with the enzymes PRORP, ELAC or TRNT1. This allows a structural snapshot of the methylation, 5'- and 3'-processing, as well as 3'-CCA addition steps of mt-tRNA maturation, and provide structural snapshots explaining how mt-tRNA can be sequentially processed.

Almost all the hypothesis derived from the structural observations are backed up by in vitro functional analyses, so this work is a great example illustrating how powerful integrated structure-function approaches are to characterize molecular mechanisms. Considering the substantial amount of work, the elegant combination of challenging techniques and the novelty of the findings presented herein, this manuscript is deserving publication in Nature Communications. This is contingent upon addressing a few points for enhanced accuracy, and incorporating more didactic elements for a non-specialist readership.

One of the major point is the interpretation and model building of the less well resolved parts of the cryo-EM maps. According to the supplementary table and figures, the structures of PRORP, ELAC and TRNT1 were solved up to respective resolutions of 4.5, 4.0 and 4.5 Å. These are often not high enough to allow a precise positioning of side chains of aminoacids, nucleotide bases or ions. Here, it seems to be the case for figure panels 3d and 3e, 4c and 5d, in which the displayed densities appear bulky and could accommodate a lot of different things. As it is not mentioned in the material and methods sections, could the authors try to improve the sharpening of their cryo-EM maps, by using local sharpening tools such as DeepEMhancer or LocScale? This might help for model building and visualization of moieties of interest.

For instance, on p.10 the authors claim that "Nucleotides G1 and A2 of tRNA Ser (AGY) appear unpaired in the density map and are buried together with nucleotide C73 of tRNA^{His} in the ELAC2 active site". While this may indeed be true, to illustrate this idea, Figure 4d shows a broad and continuous density around nucleotides A2 and G1 of tRNA^{Ser}, as well as C73 of tRNA^{His}. However, no density is shown for C70 and the 3' end of tRNA^{Ser}. Could the authors display the cryo-EM map around the latter, or at least clarify whether there is no density for the 3'-end of tRNA^{Ser}?

Figure 1c, left panel: The TRMT10C/SDR5C1 complex has already been shown to methylate mt-tRNAs in other publications. The methyltransferase activity assays shown in Figure 1c (left panel) suggest that this is also the case for the synthetic pre-tRNA chosen, since the radioactivity levels (monitoring H3-CH3 incorporation when incubating in vitro TRMT10C/SDR5C1 with synthetic pre-tRNAs) are higher than in a blank condition, described in the legend as "assay without tRNA". This suggests that the TRMT10C/SDR5C1 complex does not harbor an auto-methyltransferase activity, but the conclusion that "methylation assays [...] show that TRMT10C/SDR5C1 complex methylates both pre-tRNA^{Ile} and pre-tRNA^{His-Ser}" cannot be drawn from these experiments. Could the authors add a control condition with synthetic pre-mt-tRNA and without TRMT10C/SDR5C1, which would show the (hopefully null) basal incorporation level of H3-CH3 moiety to synthetic mt-tRNA without the multienzymatic complex? This condition is suggested to have been performed in the Material & Methods section, please clarify what is actually shown on the figures. Also, please state the number of repeats of an experiment when displaying its quantification with error bars.

Image processing schemes and FSC graphs : in Suppl Figure 2, Legend of panel f reads "Fourier shell correlation with and without solvent masks (maps 1 and 2 respectively)". It is not totally clear which 3D reconstruction or mask this refers to, there are several maps presented in panel c. Isn't it the gold-standard FSC of maps 1 and 2 (global and focused refinements) shown in panel c, instead? The same imprecision between figure panel and legend occurs in Suppl. Figures 3, 4 and 5.

Overall, the manuscript is straightforward and understandable. However, it could be made more accessible to a broader audience by providing better guidance through the presentation of the 3D structures.

Firstly, in many places in the results section, the text provides precise details about domains/amino acids of interest which are difficult to visualize in the figures. Please add labels and visual hints on the figures showing where to look at:

- p.8, in the depiction of the structure of RNase P processing tRNA part, the authors refer to a 3.10 helix and loops of PRORP which contact nucleotides U-1 and U-2 and C-3 of tRNA^{His} leader: please label residues 466-450 forming the 3.10 helix on figure 3b, as a naive reader cannot clearly see where it is. Similarly, the interaction between TRMT10C NTD (61-106) and the T-loop of tRNA described on the last paragraph p.8 is a bit hard to localize on Figure 3a for a non-specialist reader.
- The color coding between Figure 3a and b is also somehow confusing (green is for TRMT10C on panel 3a, and the same color is used on panel 3b, which is a close-up on a region of panel 3a, for a loop of PRORP interacting with the 5' leader of tRNA^{His}).
- Along the same line, the active site of ELAC2 mentioned in p.9 is difficult to see on Figure 4a, it might help to add labels on this panel or another panel zooming on the region of interest.
- Figure 5f and p.13, interactions between TRNT1 and tRNA: please add labels showing G19-C56 base pair, D- and T-loop on panel 5f

Typo / text corrections:

p.9 – comparison of atomic models of ELAC2 and ELAC1 is shown on Figure 4b, not 4a. Please modify the sentence : "(...)-lactD) domain of ELAC2 has extensions from the core that are not present in ELAC1 and that are used to bind pre-tRNAs (Fig. 4a)" accordingly.

p.10 – Regarding ELAC2, which would be "primed for cleavage", the text refers to molecular details "the catalytic residues in position to coordinate two metal ions required for endonucleolytic cleavage and the scissile phosphodiester bond positioned in the active site (Fig. 4b)" which are not distinguishable on this panel. Do the authors mean 4d, instead?

Reviewer #3 (Remarks to the Author):

Review of the manuscript Nature Comm by Meynier et al.:

In this work, the authors reveal novel insight into the maturation processes of human mitochondrial tRNAs. This process is catalysed by RNase P and ELAC2 (RNase Z L) that respectively process the 5'-leader and 3'-trailer regions of the pre-tRNA. Other key components of the mitochondrial pre-tRNA maturation are: the methyltransferase TRMT10C – responsible for N1-methylation of purine 9 (A/G)-; the 3'-CCA adding enzyme named TRNT1; and the dehydrogenase SDR5C1.

Although previous studies had provided some information about the role of each protein, how these different elements assemble into the mitochondrial pre-tRNA maturation machinery and orchestrate the different steps, remained highly elusive. Moreover, some mitochondrial pre-tRNA transcripts are organised in a way that the 3'-trailer is directly fused to the 5'-leader of a different pre-tRNA, and thus, the cleavage of the 3'-trailer by ELAC2 would yield a mature 5'-end of the adjacent pre-tRNA. However, the structural details as well as a rationale for the order of the different mitochondrial tRNA maturation processes were not well known.

The high resolution cryo-EM structures reported in this study provide unprecedented details on all these important processes for mitochondrial tRNA maturation. Important novel aspects of the study are:

- Recognition of the pre-tRNA substrate and methylation of the N1 in the purine base position 9 by the methyl-transferase TRMT10C.
- An explanation for the sequential processing of the 5' leader and 3' trailer of pre-tRNA^{His-Ser} by PRORP and ELAC2 respectively.
- Addition of the 3'CCA to the end of the tRNA acceptor stem by TRNT1.
- A scaffolding role of the complex formed by TRMT10C/SDR5C1 that supports the different mitochondrial pre-tRNA maturation steps.
- Unprecedented eukaryotic structures of ELAC2 and TRNT1 bound to pre-tRNA substrates.
- Unprecedented structures of human TRMT10C with SAH, providing novel aspects for the methylation reaction.
- The different structures and biochemical experiments provide a rationale for the sequential order used by the different enzymes during mitochondrial tRNA maturation.

Overall, this is a remarkable study by Carine Tisé and Ben Luisi's Labs that substantially contributes to advance our knowledge on mitochondrial tRNA maturation. At the same time, some of the results are likely applicable to the processing of some cytosolic tRNAs. The manuscript is written in a very clear and elegant way and the interpretation of the results and discussion is well justified. Thus, I fully recommend publication of this article, which would be of high interest for the broad readership of this journal.

I only have few suggestions:

- Page 4: perhaps a bit clearer nomenclature, or explanation, could be given at the beginning to describe the meaning of things in parenthesis in the superscripts of pre-tRNA^{His} (0,Ser) and pre-tRNA^{Ile} (0,4). For example, the first pre-tRNA lacks the 5'-leader, which presumably corresponds to "0", but what does it mean there "Ser" ? Maybe a scheme like in Fig. 1 a could be used to illustrate these tRNA controls, which could be shown in the Supp. Figures.
- Page 17, Discussion: the proposed model of tRNA quality-control by blocking different maturation enzymes (PRORP, ELAC2, TRNT1) is based on complexes with a rather special pre-tRNA transcript

(fusion of pre-tRNA^{His}-tRNA^{Ser}). How this model would be compatible with other “canonical” mitochondrial tRNAs that are not directly fusing two pre-tRNAs? And with other tRNAs, which unlike tRNA^{Ser} (AGY) contain a D-stem?

- The role of SDR5C5 in the different complexes is perhaps the less described aspect of this study. The EM structures support tetramerization of SDR5C5 in the different complexes. However, what would be the rationale for such oligomerization in the context of a likely situation where two pre-tRNAs can be simultaneously processed within the whole complex? This is briefly mentioned in the discussion, page 18, but a bit more developed explanation would be nice.

- Page 3, : “Conflicting results...” change to “distinct results...”

- Page 3: “we reconstituted.....ELAC2 or TRMT10C enzymes with the pre-tRNA bound to...” change “the pre-tRNA” to “a pre-tRNA...”

- Page 4: “amino-acylated”.....please remove hyphen. Aminoacylation is used without hyphen in other parts of the text.

- Figure 1 panel b: to facilitate the reading of the names of domains in the architecture scheme of ELAC2, the authors could use for example white font. The black font within the Beta-lactamase boxes are difficult to read (at least in the PDF I download).

- Supp. Figure 1 panel a. The details about how the pre-tRNA was stained, the expected MW (pointing arrow missing) and the presence of several pre-tRNA^{Ile} bands (of high and low molecular weight?) should be discussed.

- Supp. Figure 1 panel b: a plus symbol might be written before the cofactor/analogues names as the T_m are calculated for the complexes (not for the small molecules alone). This applies to green and beige bars....

- Supp. Figure 6: to avoid confusion, and if this is correct, please label MRPP3 as PRORP.

- figure 3 panel b: the green color used to highlight the loops in PRORP is confusing as these are similar to TRMT10C (also in green). I suggest to recolor these loops.

- The scale-up of Fig. 3 panel f, would help to see better the influence of TRMT10C on the activity of PRORP.

- The HxHxDH motif that coordinates the catalytic Zn ions should be highlighted in Supp. Fig 8a. This seems to be around position 550 in human ELAC2?

- Supp. Fig. 8b: there seems to be a decrease of on the band corresponding to tRNA^{Ser} for some deletion mutants of TRMT10C. Thus, is the statement in page 10 supported by this experiment? Or is there another evidence supporting that the interaction TRMT10C – ELAC2 is not important for tRNA processing? Given the proposed scaffolding role of complexes formed by TRMT10C to support processing activity, one could argue that no impact on the activity upon weakening the interaction by TRMT10C (mutants) is controversial.

- Page 12: “...this interaction pattern is identical to that made by cytosine in Watson-Crick base pairing...” replace identical by “equivalent”.

- Supp. Fig. 11 Legend: to avoid confusion, please replace MRPP1 by TRMT10C.

- PDB and EMD codes of the final deposited complexes generated in this work are lacking in Supp. Table 1.

Reviewer:

Andrés Palencia

National Institute of Health and Medical Research (Inserm)

Institute for Advanced Biosciences (IAB), Grenoble, Inserm-UGA-CNRS

Group Leader of Structural Biology of Novel Drug Targets in Human Diseases

Webpage: <https://noveltargets-palencia.com>

Manuscript : NCOMMS-23-57981

We thank the three reviewers for their constructive comments, and we have made extensive changes to the manuscript in light of their comments. We are happy to submit a revised version of our manuscript and hope that it is now suitable for publication in *Nature Communications*.

Below, we address each of the reviewers' comments.

Dr Carine Tisné

REVIEWER COMMENTS

Reviewer #1 (Remarks to the Author):

In mammalian mtDNA, tRNA genes are encoded between rRNA and mRNA genes without any intervening sequence. Once tRNAs are processed by RNases P and ELAC2 from the long precursor RNAs transcribed from mtDNA, rRNAs and mRNAs are generated. Then, the tRNAs undergo 3'-CCA addition by TRNT1. During tRNA processing, a wide variety of post-transcriptional modifications are introduced. Unique to metazoan mitochondria, RNase P is an RNA free multi-enzyme complex composed of PRORP, TRMT10C and SDR5C1. In this study, authors solved several cryoEM structures of precursor tRNAs bound by these maturation enzymes. The complex of pre-tRNAs bound by TRMT10C/SDR5C1 and PRORPP are already reported by another group a few years ago. So, the originality of this study is two complex structures of pre-tRNA-TRMT10C/SDR5C1 together with ELAC2 or TRNT1. This study surely contributes to our deep understanding of tRNA maturation process in human mitochondria. This reviewer has several comments and concerns that need to be addressed before publication.

Since PRORP actually recognizes tRNA and TRMT10C, 5' processing of tRNA should coordinate with m9R methylation. However, this reviewer does not see any clear interaction between ELACS and TRMT10C, and TRNT1 and TRMT10C, indicating that 3' processing and CCA addition take place independent of m9R methylation. Because various tRNA-modifying enzymes are involved in tRNA maturation in mitochondria, it is quite natural to think other tRNA-modifying enzymes also recognize tRNA precursors instead of TRMT10C/SDR5C1 binding when their 3' ends are processed. Authors should mention and properly discuss other pathways for tRNA maturation.

⇒ We have added the following sentence in the discussion to mention that the TRMT10C/SDR5C1 complex is not required for further steps of nucleotide modifications: «For the remaining steps of tRNA maturation, such as the incorporation of nucleotide chemical modifications, the TRMT10C/SDR5C1 complex is not required. »

In terms of cryoEM maps, the lower parts of complexes (SDR5C1, TRMT10C, tRNA anticodon loop) show high resolution. In contrast, the upper part (tRNA Ser, PROPP, ELAC2, TRNT1) show relatively low resolution (over 4Å). Authors cannot discuss anything about the side chain position of the specific residues throughout the manuscript.

⇒ We agree that the resolution in the upper part of the complexes is lower. In the revised manuscript, we describe the interaction by giving the type of secondary structures (helix, loop...) with the numbers of the residues forming these secondary structures involved in the interactions. In the two cases where it was interesting to mention that a specific interaction could occur, we have added the term « likely ».

p. 9 : «For instance, H447 is likely π -stacked on U-2.»

P. 13 : «a molecule of CDP that is likely π -stacked with the substrate A73 nucleotide»

Because these maps are 2.7-3.2Å resolution, this reviewer supposes H₂O should not be modelled. H₂O can be modelled if resolution is lower than 2-2.2Å resolution. In addition, this reviewer supposes several Mg ions exist around tRNA (for example, near A9 in sample2), it's better to model these Mg ions first.

⇒ We thank the reviewer for this comment. We have checked the H₂O molecules present in our structures of complexes 1 and 2 and all have been removed and the updated pdb re-deposited to the PDB accordingly. Mg ions are probably present near A9, but when we tried to model Mg ions there, the online validation tools or in Coot did not give scores that confirm their presence with confidence. We have therefore decided not to add them.

Local resolution maps of sample2-4 maps (as seen in Sup Fig1E) are required, because overall resolution of refined maps (showed in Sup Fig345D) is not consistent with local resolution of each region.

⇒ We have generated local resolution maps and added figure panels with maps coloured by local resolution to the Supplemental Figures included in the revised manuscript.

Page 2 (3rd paragraph): N¹-methylation. "N" should be italic and "1" should be superscript.

⇒ This has been changed.

Page 3 (2nd paragraph): "significantly" should be "significant"

⇒ This has been changed.

Page 5, lines 1-4: It's useful for readers to describe the constructs of recombinant proteins whether they have MTS or not.

⇒ A sentence has been added p. 5: «All proteins were expressed without the mitochondrial transport signal (MTS). »

Page 5, line 5: Appropriate references are necessary for this sentence, ”

⇒ References have been added.

Page 5, line 7: An explanation should be included for notation of His(0, Ser) and Ile(0,4) in the pre-tRNA^{His(0, Ser)} and pre-tRNA^{Ile(0,4)} respectively.

⇒ A sentence has been added p. 5 to explain the nomenclature and we have added a Supplementary Fig. 1a with all the tRNA used in this study and to explain the numbering system: "In pre-tRNA^{His(0, Ser)} and pre-tRNA^{Ile(0,4)}, the 0 corresponds to 0 nucleotide in the 5'-leader and, Ser and 4 corresponds to tRNA^{Ser}(AGY) and 4 nucleotides as 3'-trailer, respectively.»

⇒ A sentence has also been added in the figure legend of Fig. 1 : «The sequence of the other pre-tRNAs used in this study and the nucleotide numbering used in the atomic coordinates deposited in the PDB are shown in Supplementary Fig. 1a.»

Page 7: It is better to clearly mention the region of ELAC2 that clashes with pre-tRNA 5'-leader in the main text. In addition, the reviewer would like the authors to check if the region is conserved in other organisms, aiming to validate the generality of their findings.

⇒ A sentence has been added p. 7: «These clashes occur at the level of the C-terminal helix of ELAC2 (residues 761-794) that is conserved in metazoans, present in *Saccharomyces cerevisiae*, but not in *Schizosaccharomyces pombe* and not conserved in plants (Supplementary Fig. 8a).»

Page 8 Line 8: "G1 and U2 of tRNAHis..." should be "G1 and U-1 of tRNAHis..."

⇒ No, this is correct, the loops containing D499 and D503 (in red in Fig. 2b, visible in Fig. 3c) are in contact with G1 and U2.

Page 8, line 10: Explain "310 helix".

⇒ We have added p. 8 that a « 3_{10} helix is a helix with three residues per turn.»

Page 8: the second paragraph: The authors should use figures to illustrate what is written here.

⇒ A Supplementary Fig. 7b has been added to illustrate the interaction between the TRMT10C MTase domain and PRORP.

Page 8: Regarding the results in Fig 3f: The delta(1-80) mutant appears to be active, and the authors should explain why. The authors should also comment in the main text on the results of the delta(1-166) mutant.

⇒ We have changed the text to better explain the results of the gel in Fig. 3f, and the new sentence reads as follows :

«Deletion of this helix $\alpha 1$ has no apparent effect on PRORP activity, whereas once the two first helices of TRMT10C are deleted, *i.e.* in the TRMT10C Δ (1-106) and TRMT10C Δ (1-166) mutants, the TRMT10C/SDR5C1 complex is no longer able to promote PRORP-mediated cleavage of pre-tRNA^{His-Ser} (Fig. 3f).»

Page 10 line 2: Fig. 4b should be 4c?

⇒ Fig. 4b has been replaced by Fig. 4 a,c.

Page 10 (1st paragraph): no figure for surface charge in Fig. 4a?

⇒ Fig. 4d has been added with the surface charge for ELAC2.

Page 10 Last paragraph: Replacing of pre-tRNA with cytoplasmic tRNA from other model is not impressive, because ELAC2 structure would be changed by tRNA substrate. Especially, the position of protrusion in the Nter β -lactD would be easily changed. This reviewer does not understand what the authors intend to describe with this model in Figure 5g. It should be removed or relocated to SI figure.

⇒ This paragraph and the figure have been moved to discussion to discuss this point together with TRNT1.

Page 12 Line19: ATP accommodation model should be shown in the figure.

⇒ We have added a Supplementary Fig. 9c to show the ATP accommodation model.

Page 13: "The conserved residue L180 interacts with G34 that is further stabilized by chain C of SDR5C1 via residues (98-105) from a lysine-rich loop that also interacts with U32." But, SDR5C1 is not shown in Fig 6.

⇒ A Supplementary Fig. 10b has been added to show this.

Page 15: "For the human TRMT10A and the *S. cerevisiae* Trm10 enzymes..." Provide the AlphaFold models of these enzymes (AF-Q12400 and AF-Q8TBZ6) in SI figure.

⇒ Supplementary Fig. 11d has been added to describe this point.

Page 16: Based on SAH model, SAM model can be generated to explain the reaction mechanism mediated by some specific residues.

⇒ The sentence has been modified as requested.

Page 17, last sentence: Many pathogenic mutations have been reported in human mitochondrial tRNA. This reviewer asks the authors to mention whether any of the pre-tRNA bases involved in the interactions with each protein identified in this paper are related to pathogenic mutations. Also, it is better to mention whether pathogenic mutations have been reported in the proteins as well.

⇒ A paragraph in Discussion and a Supplementary Fig. 13 have been added.

Page 18, lines 3-5: The authors should explain exactly what kind of transcripts they are referring to here as non-canonical mitochondrial pre-tRNA.

⇒ « Non-canonical » has been changed by «structurally-fragile» mitochondrial pre-tRNA.

Page 18: Besides His-Ser pair, it is interesting for this reviewer to discuss whether this character is preserved or not in the other mt-tRNA clusters?

⇒ A sentence has been added in the discussion section about the second cluster of mt-tRNAs, but less information is available about the processing of this cluster (order and which enzyme).

«A second cluster of tRNAs is found on the mt-DNA light-strand (tRNA^{Tyr}-tRNA^{Cys}-tRNA^{Asn}-tRNA^{Ala}). TRMT10C/SDR5C1 likely binds to each tRNA and further studies will be needed to determine the chronology of processing events and which enzyme, either PRORP or ELAC2, cleaves which junction.»

Fig.1c: Pre-tRNA^{His-Ser} should be Pre-tRNA^{His(0,Ser)}

⇒ This has been corrected.

Fig. 2b: the presence of 5'-leader prevents cleavage by ELAC2. The cleavage activity of ELAC2 for each tRNA should be quantified by the band intensity of the product.

⇒ The quantification of the band intensity of the product shows that less than 20% of the product is observed when the 5'-leader is present compared to 100% or 50% of product for pre-tRNA^{Ile(0,4)} and pre-tRNA^{His(0,Ser)} respectively. The text has been changed p. 8 to now read: « the quantification of the band intensity of the cleavage product by ELAC2 (Fig. 2b) shows that only 20% of the cleavage products are observed when the 5'-leader is present on the pre-tRNAs, compared to 100% or 50% of cleavage products for pre-tRNA^{Ile(0,4)} and pre-tRNA^{His(0,Ser)}, respectively, with no 5'-leader. ELAC2 cleavage activity is thus inhibited by the presence of the 5'-leader in pre-tRNA^{Ile} and pre-tRNA^{His-Ser} (Fig. 2b).»

Fig.3c: A479 should be D479A

⇒ This has been corrected.

Fig. 3f: The line with "Pre-tRNA^{His}(0,Ser)" is overlapped with the gel image (Ref Lane). Please improve.

⇒ This has been corrected.

Fig. 5c: The density of CDP is not clearly visualized. It's better to show this figure with different angle. If the density of phosphate group is observed, discuss binding residues with phosphate group.

⇒ We have changed Fig. 5c as requested. The density of phosphate group is not observable in the density map.

Sample 2 and 3. It is confusing that model numbering and Figure legend are different at tRNAs. Mg1 and Zn1 densities are not shown in there maps. Please move this part to discussion section.

⇒ We have added a Supplementary Fig. 1a to show to show the different tRNAs used in this study and the numbering used in the atomic coordinates. It was important to keep the numbering used for tRNAs and known by the RNA community to immediately identify the part of the tRNA with the numbering. For instance, the anticodon nucleotides with this nomenclature is always 34, 35 and 36.

⇒ We have kept Mg1 and Zn1 because we observed a residual density at the position where these ions are expected.

Sample4. The densities of CDP and surrounding residues of TRNT1 are weak, and binding mode against CDP is not clearly defined. You should move this part in discussion section.

⇒ We have added one view for Fig. 5c and the ATP accommodation in Supplementary Fig. 9c as requested by the reviewer. We have changed the text to take into account that the low resolution of the map in this part : «a molecule of CDP that is likely π -stacked with the substrate A73 nucleotide.»

Reviewer #2 (Remarks to the Author):

In this manuscript, V. Meynier and co-workers. have combined structural and functional approaches to give a comprehensive description of the machinery required for the maturation of human mitochondrial tRNA. In this organelle, precursors to mt-tRNA are matured from polycistronic transcripts by a series of events, namely endonucleolytic cleavages, addition of a trinucleotide moiety at their 3' end (3'-CCA), as well as chemical modifications such as methylations. These maturation events are performed through the association of pre-mt-tRNA to a maturation platform composed of proteins TRMT10C and SDR5C1, with which the enzymes PRORP, ELAC and TRNT1 will associate for 5'-, 3'-endonucleolytic cleavage and 3'-CCA addition, respectively. Although structures of mt-RnaseP (TRMT10C/SDR5C1/PROP) complex bound to pre-tRNA^{Tyr} have recently been published, little is known regarding how the rest of the tRNA maturation sequence happens. The authors have thus used cryo-electron microscopy to propose four atomic 3D structures of maturing pre-mt-tRNA bound to the TRMT10C/SDR5C1 platform alone or associated with the enzymes PRORP, ELAC or TRNT1. This allows a structural snapshot of the methylation, 5'- and 3'-processing, as well as 3'-CCA addition steps of mt-tRNA maturation, and provide structural snapshots explaining how mt-tRNA can be sequentially processed.

Almost all the hypothesis derived from the structural observations are backed up by in vitro functional analyses, so this work is a great example illustrating how powerful integrated structure-function approaches are to characterize molecular mechanisms. Considering the substantial amount of work, the elegant combination of challenging techniques and the novelty of the findings presented herein, this manuscript is deserving publication in Nature Communications. This is contingent upon addressing a few points for enhanced accuracy, and incorporating more didactic elements for a non-specialist readership.

One of the major point is the interpretation and model building of the less well resolved parts of the cryo-EM maps. According to the supplementary table and figures, the structures of PRORP, ELAC and TRNT1 were solved up to respective resolutions of 4.5, 4.0 and 4.5 Å. These are often not high enough to allow a precise positioning of side chains of aminoacids, nucleotide bases or ions. Here, it seems to be the case for figure panels 3d and 3e, 4c and 5d, in which the displayed densities appear bulky and could accommodate a lot of different things. As it is not mentioned in the material and methods sections, could the authors try to improve the sharpening of their cryo-EM maps, by using local sharpening tools such as DeepEMhancer or LocScale? This might help for model building and visualization of moieties of interest.

⇒ We tested various post-processing/sharpening for the maps presented in the manuscript (including EMhancer). In our evaluations, the maps deposited provided the highest quality results.

For instance, on p.10 the authors claim that “Nucleotides G1 and A2 of tRNA Ser (AGY) appear unpaired in the density map and are buried together with nucleotide C73 of tRNA^{His} in the ELAC2 active site”. While this may indeed be true, to illustrate this idea, Figure 4d shows a broad and continuous density around nucleotides A2 and G1 of tRNA^{Ser}, as well as C73 of tRNA^{His}. However, no density is shown for C70 and the 3' end of tRNA^{Ser}. Could the authors display the cryo-EM map around the latter, or at least clarify whether there is no density for the 3'-end of tRNA^{Ser}?

⇒ Fig. 4e has been changed to show the density around C70.

Figure 1c, left panel: The TRMT10C/SDR5C1 complex has already been shown to methylate mt-tRNAs in other publications. The methyltransferase activity assays shown in Figure 1c (left panel) suggest that this is also the case for the synthetic pre-tRNA chosen, since the radioactivity levels (monitoring H3-CH3 incorporation when incubating in vitro TRMT10C/SDR5C1 with synthetic pre-tRNAs) are higher than in a blank condition, described in the legend as “assay without tRNA”. This suggests that the TRMT10C/SDR5C1 complex does not harbor an auto-methyltransferase activity, but the conclusion that “methylation assays [...] show that TRMT10C/SDR5C1 complex methylates both pre-tRNA^{Ile} and pre-tRNA^{His-Ser}” cannot be drawn from these experiments. Could the authors add a control condition with synthetic pre-mt-tRNA and without TRMT10C/SDR5C1, which would show the (hopefully null) basal incorporation level of H3-CH3 moiety to synthetic mt-tRNA without the multienzymatic complex? This condition is suggested to have been performed in the Material & Methods section, please clarify what is actually shown on the figures. Also, please state the number of repeats of an experiment when displaying its quantification with error bars.

⇒ The control condition was the pre-mt-tRNA without TRMT10C/SDR5C1 as described in Materials and Methods. We have clarified this point in the Fig. 1 legend. All methyltransferase assays were repeated three times and Fig. 1c has been changed to show the three points together with the error bars, and it is now stated in the Materials and Methods section that the experiments were done in triplicate.

Image processing schemes and FSC graphs : in Suppl Figure 2, Legend of panel f reads “Fourier shell correlation with and without solvent masks (maps 1 and 2 respectively)”. It is not totally clear which 3D reconstruction or mask this refers to, there are several maps presented in panel c. Isn't it the gold-standard FSC of maps 1 and 2 (global and focused refinements) shown in panel c, instead? The same imprecision between figure panel and legend occurs in Suppl. Figures 3, 4 and 5.

⇒ We thank the reviewer for noticing this imprecision and have revised the figure legends to make these more clear.

Overall, the manuscript is straightforward and understandable. However, it could be made more accessible to a broader audience by providing better guidance through the presentation of the 3D structures.

Firstly, in many places in the results section, the text provides precise details about domains/amino acids of interest which are difficult to visualize in the figures. Please add labels and visual hints on the figures showing where to look at:

- p.8, in the depiction of the structure of Rnase P processing tRNA part, the authors refer to a 3.10 helix and loops of PRORP which contact nucleotides U-1 and U-2 and C-3 of tRNA^{His} leader: please label residues 466-450 forming the 3.10 helix on figure 3b, as a naive reader cannot clearly see where it is. Similarly, the interaction between TRMT10C NTD (61-106) and the T-loop of tRNA described on the last paragraph p.8 is a bit hard to localize on Figure 3a for a non-specialist reader.

The color coding between Figure 3a and b is also somehow confusing (green is for TRMT10C on panel 3a, and the same color is used on panel 3b, which is a close-up on a region of panel 3a, for a loop of PRORP interacting with the 5' leader of tRNA^{His}).

⇒ The 3¹⁰ helix (res 446-449) has been labeled in Fig. 3b. The T-loop and the anticodon loops have been labeled in Fig. 3A. The color of the loop in green in Fig. 3b is now in pink.

- Along the same line, the active site of ELAC2 mentioned in p.9 is difficult to see on Figure 4a, it might help to add labels on this panel or another panel zooming on the region of interest.

⇒ Fig. 4 has been changed and labels have been added to better describe the active site of ELAC2.

- Figure 5f and p.13, interactions between TRNT1 and tRNA: please add labels showing G19-C56 base pair, D- and T-loop on panel 5f

⇒ Fig. 5f has been moved to Fig. 6b and labels have been added.

Typo / text corrections:

p.9 – comparison of atomic models of ELAC2 and ELAC1 is shown on Figure 4b, not 4a. Please modify the sentence : “(...)b-lactD) domain of ELAC2 has extensions from the core that are not present in ELAC1 and that are used to bind pre-tRNAs (Fig. 4a)” accordingly.

⇒ This has been corrected.

p.10 – Regarding ELAC2, which would be “primed for cleavage”, the text refers to molecular details “the catalytic residues in position to coordinate two metal ions required for endonucleolytic cleavage and the scissile phosphodiester bond positioned in the active site (Fig. 4b)” which are not distinguishable on this panel. Do the authors mean 4d, instead?

⇒ We have replaced Fig. 4b by Fig. 4 a,c.

Reviewer #3 (Remarks to the Author):

Review of the manuscript Nature Comm by Meynier et al.: In this work, the authors reveal novel insight into the maturation processes of human mitochondrial tRNAs. This process is catalysed by RNase P and ELAC2 (RNase Z L) that respectively process the 5' leader and 3' trailer regions of the pre-tRNA. Other key components of the mitochondrial pre-tRNA maturation are: the methyltransferase TRMT10C – responsible for N1-methylation of purine 9 (A/G)–; the 3'-CCA adding enzyme named TRNT1; and the dehydrogenase SDR5C1.

Although previous studies had provided some information about the role of each protein, how these different elements assemble into the mitochondrial pre-tRNA maturation machinery and orchestrate the different steps, remained highly elusive. Moreover, some mitochondrial pre-tRNA transcripts are organised in a way that the 3'-trailer is directly fused to the 5'-leader of a different pre-tRNA, and thus, the cleavage of the 3'-trailer by ELAC2 would yield a mature 5'-end of the adjacent pre-tRNA. However, the structural details as well as a rationale for the order of the different mitochondrial tRNA maturation processes were not well known.

The high resolution cryo-EM structures reported in this study provide unprecedented details on all these important processes for mitochondrial tRNA maturation. Important novel aspects of the study are:

- Recognition of the pre-tRNA substrate and methylation of the N1 in the purine base position 9 by the methyl-transferase TRMT10C.
- An explanation for the sequential processing of the 5' leader and 3' trailer of pre-tRNA^{His-Ser} by PRORP and ELAC2 respectively.
- Addition of the 3'CCA to the end of the tRNA acceptor stem by TRNT1.
- A scaffolding role of the complex formed by TRMT10C/SDR5C1 that supports the different mitochondrial pre-tRNA maturation steps.
- Unprecedented eukaryotic structures of ELAC2 and TRNT1 bound to pre-tRNA substrates.
- Unprecedented structures of human TRMT10C with SAH, providing novel aspects for the methylation reaction.
- The different structures and biochemical experiments provide a rationale for the sequential order used by the different enzymes during mitochondrial tRNA maturation.

Overall, this is a remarkable study by Carine Tisné and Ben Luisi's Labs that substantially contributes to advance our knowledge on mitochondrial tRNA maturation. At the same time, some of the results are likely applicable to the processing of some cytosolic tRNAs. The manuscript is written in a very clear and elegant way and the interpretation of the results and discussion is well justified. Thus, I fully recommend publication of this article, which would be of high interest for the broad readership of this journal.

I only have few suggestions:

- Page 4: perhaps a bit clearer nomenclature, or explanation, could be given at the beginning to describe the meaning of things in parenthesis in the superscripts of pre-tRNA^{His} (0,Ser) and pre-tRNA^{Ile} (0,4). For example, the first pre-tRNA lacks the 5'-leader, which presumably corresponds to "0", but what does it mean there "Ser" ? Maybe a scheme like in Fig. 1 a could be used to illustrate these tRNA controls, which could be shown in the Supp. Figures.

⇒ A sentence has been added p. 5 to explain the nomenclature : «In pre-tRNA^{His(0,Ser)} and pre-tRNA^{Ile(0,4)}, the 0 corresponds to 0 nucleotide in the 5'-leader and, Ser and 4 corresponds to tRNA^{Ser}(AGY) and 4 nucleotides as 3'-trailer, respectively.»

⇒ We have also added a Supplementary Fig. 1a to show the secondary structure of all tRNAs used in this study.

- Page 17, Discussion: the proposed model of tRNA quality-control by blocking different maturation enzymes (PRORP, ELAC2, TRNT1) is based on complexes with a rather special pre-tRNA transcript (fusion of pre-tRNA^{His}-tRNA^{Ser}). How this model would be compatible with other “canonical” mitochondrial tRNAs that are not directly fusing two pre-tRNAs? And with other tRNAs, which unlike tRNA^{Ser} (AGY) contain a D-stem?

⇒ We have added a Supplementary Fig. 13 to show how a more canonical pre-tRNA with a more stable elbow can be bound in the different tRNA maturation complexes solved in this paper.

This is now discussed p. 21-22: «tRNA^{Lys3}, which has a stable elbow, can also be placed in the complex of mitochondrial RNase P, Z and 3'-CCA addition described in this paper (Supplementary Fig. 13). This shows how more canonical mitochondrial pre-tRNAs, with more stable elbows, can be bound by TRMT10C/SDR5C1 complex for 5'- and 3'-processing and 3'-CCA addition.»

- The role of SDR5C1 in the different complexes is perhaps the less described aspect of this study. The EM structures support tetramerization of SDR5C1 in the different complexes. However, what would be the rationale for such oligomerization in the context of a likely situation where two pre-tRNAs can be simultaneously processed within the whole complex? This is briefly mentioned in the discussion, page 18, but a bit more developed explanation would be nice.

⇒ This is an interesting point, and one possibility is that the tetrameric quaternary organisation has cooperative behaviour ; however, we are not aware of any data that indicates an interplay between two simultaneously bound substrates. We have added a sentence in the discussion to explain that SDR5C1 tetramer allows to bind two TRMT10C-bound pre-tRNA, one on each tetramer face. «The SDR5C1 tetramer acts as a platform that can bind two TRMT10C/pre-tRNA units, one on each face of the tetramer.»

-Page 3, : “Conflicting results...” change to “distinct results...”

⇒ This has been corrected.

-Page 3: “we reconstituted.....ELAC2 or TRNT1 enzymes with the pre-tRNA bound to...” change “the pre-tRNA” to “a pre-tRNA...”

⇒ This has been corrected.

- Page 4: “amino-acylated”.....please remove hyphen. Aminoacylation is used without hyphen in other parts of the text.

⇒ This has been corrected.

- Figure 1 panel b: to facilitate the reading of the names of domains in the architecture scheme of ELAC2, the authors could use for example white font. The black font withing the Beta-lactamase boxes are difficult to read (at least in the PDF I download).

⇒ We have changed the colours slightly in panel b of Fig. 1.

- Supp. Figure 1 panel a. The details about how the pre-tRNA was stained, the expected MW (pointing arrow missing) and the presence of several pre-tRNA^{Ile} bands (of high and low molecular weight?) should be discussed.

⇒ The expected MW for TRMT10C and SDR5C1 have been added. The bands of high and low molecular weights are already present in the sample of pre-tRNA^{Ile} alone and correspond probably to species that are not properly folded. On a denaturing urea gel, only one band is observed for pre-tRNA^{Ile}. This is now indicated in the Suppl. Fig. 1 legend.

- Supp. Figure 1 panel b: a plus symbol might be written before the cofactor/analogues names as the T_m are calculated for the complexes (not for the small molecules alone). This applies to green and beige bars....

⇒ This has been corrected.

- Supp. Figure 6: to avoid confusion, and if this is correct, please label MRPP3 as PRORP.

⇒ This has been corrected.

- figure 3 panel b: the green color used to highlight the loops in PRORP is confusing as these are similar to TRMT10C (also in green). I suggest to recolor these loops.

⇒ This has been corrected.

- The scale-up of Fig. 3 panel f, would help to see better the influence of TRMT10C on the activity of PRORP.

⇒ This has been corrected.

- The HxHxDH motif that coordinates the catalytic Zn ions should be highlighted in Supp. Fig 8a. This seems to be around position 550 in human ELAC2?

⇒ We have added in the text ⁵⁴⁶HxHxDH⁵⁵¹ instead of HxHxDH and have highlighted it in Supp. Fig 8a. Stars have been added in Suppl Fig 8a to indicate residues involved in Zn²⁺ coordination.

- Supp. Fig. 8b: there seems to be a decrease of the band corresponding to tRNA-Ser for some deletion mutants of TRMT10C. Thus, is the statement in page 10 supported by this experiment? Or is there another evidence supporting that the interaction TRMT10C – ELAC2 is not important for tRNA processing? Given the proposed scaffolding role of complexes formed by TRMT10C to support processing activity, one could argue that no impact on the activity upon weakening the interaction by TRMT10C (mutants) is controversial.

⇒ The band for tRNA^{His} has always the same intensity. So, we think that the decrease of the band corresponding to tRNA^{Ser} for some mutants is due to degradation of tRNA^{Ser} when TRMT10C is truncated. We also observe this degradation when TRMT10C is not present. This is the (-) lane on the gel. It appears that TRMT10C prevents tRNA^{Ser} from degradation. TRMT10C is not required for processing by ELAC2, but if the tRNA is still bound to TRMT10C/SDR5C1, the gel results indicates that ELAC2 can process it. In the discussion, we explain that « *In vitro*, multiple turnovers could not be observed with TRMT10C^{27,32}, and the pre-tRNA was shown to remain bound to the TRMT10C/SDR5C1 complex²⁷ ». If the pre-tRNA is not released from the

TRMT10C/SDR5C1 complex after PRORP processing, ELAC2 is not blocked and can process the pre-tRNA.

- Page 12: "...this interaction pattern is identical to that made by cytosine in Watson-Crick base pairing..." replace identical by "equivalent".

⇒ This has been corrected.

- Supp. Fig. 11 Legend: to avoid confusion, please replace MRPP1 by TRMT10C.

⇒ This has been corrected.

- PDB and EMDB codes of the final deposited complexes generated in this work are lacking in Supp. Table 1.

⇒ This has been corrected.

REVIEWERS' COMMENTS

Reviewer #1 (Remarks to the Author):

The revised version addressed most of this reviewer's comments. I have no further comment.

Reviewer #2 (Remarks to the Author):

This revised version of the manuscript from V. Meynier and co-workers shows significant improvements compared to the initial one. All my major concerns and the ones from the other reviewers have been addressed. This work is now of great quality, and could be published in Nature Communications as such.

I have one minor comment regarding the new manuscript version and the authors response to my initial remarks regarding the cryo-EM map sharpening: the authors say in their rebuttal letter that they tried different map sharpening tools already, and that the maps they present are the best they could get out of their processing. Considering the relatively low resolution (less than 4.5Å) of some part of the maps, this is not surprising. However, the authors should indicate in the Methods section as well as in the supplementary figures describing their SPA schemes which software(s) they employed for map sharpening, as it is not entirely clear as such.

Reviewer #3 (Remarks to the Author):

The authors have done an excellent job in addressing the reviewers's concerns. I have no further comments.

Point-by-point response to the reviewers' comments :

Reviewer 2 :

This revised version of the manuscript from V. Meynier and co-workers shows significant improvements compared to the initial one. All my major concerns and the ones from the other reviewers have been addressed. This work is now of great quality, and could be published in Nature Communications as such.

I have one minor comment regarding the new manuscript version and the authors response to my initial remarks regarding the cryo-EM map sharpening: the authors say in their rebuttal letter that they tried different map sharpening tools already, and that the maps they present are the best they could get out of their processing. Considering the relatively low resolution (less than 4.5Å) of some part of the maps, this is not surprising. However, the authors should indicate in the Methods section as well as in the supplementary figures describing their SPA schemes which software(s) they employed for map sharpening, as it is not entirely clear as such.

As stated in our previous response, we had tested various post-processing/sharpening for the maps presented in the manuscript (including EMhancer as the reviewer had suggested). In our evaluations, the maps deposited provided the highest quality results. The reviewer is correct to point out that we did not explicitly describe the method of sharpening we ultimately used, which was anisotropic sharpening as implemented within Phenix. We have now updated the methods section to clarify this.